# A kinase-independent function of PAK is crucial for pathogen-mediated actin remodelling

**Anthony Davidson[☯], Joe Tyler[☯], Peter Hume[ID], Vikash Singh, Vassilis Koronakis[ID]***

Department of Pathology, University of Cambridge, Cambridge, United Kingdom

☯ These authors contributed equally to this work.
* vk103@cam.ac.uk

**Data Availability Statement:** All relevant data are within the manuscript and its Supporting Information files.

**Funding:** This project was supported by the following grants: Wellcome Trust Senior

## Abstract

The p21-activated kinase (PAK) family regulate a multitude of cellular processes, including actin cytoskeleton remodelling. Numerous bacterial pathogens usurp host signalling pathways that regulate actin reorganisation in order to promote Infection. *Salmonella* and pathogenic *Escherichia coli* drive actin-dependent forced uptake and intimate attachment respectively. We demonstrate that the pathogen-driven generation of both these distinct actin structures relies on the recruitment and activation of PAK. We show that the PAK kinase domain is dispensable for this actin remodelling, which instead requires the GTPase-binding CRIB and the central poly-proline rich region. PAK interacts with and inhibits the guanine nucleotide exchange factor β-PIX, preventing it from exerting a negative effect on cytoskeleton reorganisation. This kinase-independent function of PAK may be usurped by other pathogens that modify host cytoskeleton signalling and helps us better understand how PAK functions in normal and diseased eukaryotic cells.

## Author summary

Many bacterial pathogens drive infection by remodelling the host cell actin cytoskeleton, and p21 activated kinases (PAK's) are well known to be involved in modulating actin dynamics. *Salmonella* and EPEC generate very different actin structures in order to promote either invasion into or attachment onto host cells. Here we show that both bacteria hi-jack the same PAK signalling pathway to achieve this. PAK's crucial role is independent of its kinase activity, and instead relies on its ability to interact the host guanine nucleotide exchange factor β-PIX. PAK binding negates the inhibitory effect β-PIX has on the bacterial driven-actin remodelling that permits either invasion or attachment. This is a novel kinase-independent function for PAK that is likely used by many other bacterial and viral pathogens that target host actin cytoskeleton signalling.

## Introduction

The p21 activated kinase (PAK) family of proteins are associated with numerous key cellular processes. Between them the Class I (PAK 1, 2 and 3) and Class II (PAK 4, 5 and 6) PAKs,

Investigator Award (101828/Z/13/Z) https://wellcome.org/grant-funding awarded to VK. Medical Research Council (MR/L008122/1) https://mrc.ukri.org/funding/ awarded to VK and the Cambridge Isaac Newton Trust https://www.newtontrust.cam.ac.uk/ awarded to VK. The funders had no role in study design, data collection and analysis, decision to publish, or preparation of the manuscript.

**Competing interests:** The authors have declared that no competing interests exist.

which have both distinct and overlapping functions, regulate signalling pathways such as those involved in cell motility, cell cycle, cancer progression and gene expression [1,2]. Originally identified as serine/threonine kinases, all PAKs contain a C-terminal kinase domain, and a Cdc42/Rac interactive Binding (CRIB) domain [3,4]. Class I PAKs are autoinhibited as a result of an interaction between the kinase domain and the autoinhibitory domain (AID), though whether this interaction is inter- or intramolecular remains controversial [5,6]. Whichever model is correct, inhibition is relieved upon binding of Rac1 or Cdc42 to the CRIB (which partially overlaps with the AID), resulting in the kinase domain being exposed, stimulating its activity. Auto-phosphorylation of the protein then occurs (including within both the kinase domain itself and the CRIB/AID) stabilising PAK's open active state [7]. Class I PAKs can also be activated by binding to inositol phosphates and sphingolipids [8], and there is increasing evidence that dual binding to lipids and Rac1/Cdc42 is required for full activation [9]. Class II PAK regulation appears more complex, as they are constitutively phosphorylated whilst existing in a monomeric autoinhibited state. Numerous proteins including Cdc42 appear to be able to relieve the inhibition of the Class II PAK kinase activity upon binding [10].

Historically much PAK related research has concentrated on identifying substrates of the kinase domain, and a multitude of candidates have been discovered. These include both positive and negative regulators of actin turnover such as Myosin Light Chain (MLC) [11]), Filamin A [12], ArpC1b [13] and LIM Kinase [14]. PAKs also contain poly-proline rich regions (PPRs) which interact with SH3 domain-containing proteins such as the adaptor protein Nck [15] and the guanine nucleotide exchange factor (GEF) β-PIX [16]. PAK thus act as a scaffold for numerous signalling complexes, again leading to both stimulation and inhibition of actin rearrangement. This broad set of substrates and interactors highlight the complicated role PAK plays in actin cytoskeleton remodelling.

Many bacteria drive infection by manipulation of the host actin cytoskeleton. The Gram-negative facultative intracellular pathogen *Salmonella enterica* Serovar Typhimurium (hereafter "*Salmonella*") forces its own uptake into non-phagocytic host cells, a key step in infection ultimately leading to symptoms ranging from acute gastroenteritis to more serious systemic disease [17]. To gain entry into host cells *Salmonella* employs a cocktail of injected virulence factors that remodel the actin cytoskeleton to drive uptake through a process similar to macropinocytosis [18,19]. Previous research has identified a role for PAK in the cell's nuclear response to *Salmonella* infection [20], as well as in the generation of an acute inflammatory response [21]. However, no role for PAK in bacterial uptake has been demonstrated, with the overexpression of kinase dead PAK as well as PAK containing mutated proline residues being shown to have no negative effect on invasion [20]. PAK is known to play a key role in driving micropinocytosis in healthy cells [22], and has been reported to be subverted by a range of pathogens (both viral and bacterial) in order to rearrange the host cytoskeleton [23]. Indeed, we have recently identified a key role for Class I PAKs in the intimate attachment and subsequent actin-rich pedestal formation by the extracellular pathogens Enteropathogenic and Enterohaemorrhagic *Escherichia coli* (EPEC and EHEC) [24,25]. We therefore sought to examine the involvement of PAK in the initial remodelling of the actin cytoskeleton that drives *Salmonella* uptake.

## Results

### Class I PAKs are crucial for *Salmonella* invasion

To determine whether PAK was required for the invasion of cells by *Salmonella*, Hap1 cells were treated with a range of commercially available PAK inhibitors, before being infected for 20 minutes. All of the Class I PAK inhibitors tested (AZ13705339, FRAX486, FRAX594,

FRAX1036, G5555, IPA3 and NVS-PAK1) reduced *Salmonella* invasion by 75–85% compared to DMSO or untreated controls (Fig 1A). In contrast, neither of the Class II PAK inhibitors tested (PF3758309 or GNE 2861) had any significant effect on invasion. All of the inhibitors used interfere with ATP binding within the kinase domain except IPA-3 which binds to the PAK regulatory domain blocking Rac1/Cdc42 binding [26] (summarised in S1A Fig). Similar results were observed in both MEF (S1B Fig) and HeLa cells (S1C Fig), indicating that Class I, but not Class II PAKs are involved in *Salmonella* uptake into non-phagocytic cells.

To confirm the involvement of Class I PAKs, we used PAK1 (ΔPAK1) or PAK2 (ΔPAK2) knockout Hap1 cells. Although uptake was reduced in both cell lines (Fig 1B), ΔPAK1 cells saw the greater decrease (62%) compared to that in ΔPAK2 cells (37% decrease). Additional knockdown using siRNA (S1D Fig) of the PAK isoform not knocked out, (i.e. ΔPAK1 cells + PAK2 siRNA) resulted in invasion being impeded by approximately 75% (similar to the impairment seen in Class I PAK inhibitor treated cells). Therefore, *Salmonella* invasion involves both PAK1 and 2, with PAK1 seemingly making the greater contribution.

We have previously shown that EPEC also hijacks Class I PAKs, but unlike EPEC, which has the effector protein EspG [27], *Salmonella* has no effector known to directly interact with PAK [28]. We hypothesised that *Salmonella* could recruit PAK indirectly via injection of the effectors SopE1 and SopE2, which are able to activate the small GTPases Rac1 and Cdc42 [29], both potent activators of PAK. When ΔPAK1 Hap1 cells were transfected with Emerald-tagged PAK1 (Em-PAK1) prior to infection, Em-PAK1 was enriched at WT, but not ΔSopE1/E2, *Salmonella* entry foci (Fig 1C). Em-PAK1 was also recruited to sites of *Salmonella* entry in MEF and HeLa cells (S1E Fig). Em-PAK1 successfully restored invasion in ΔPAK1 cells infected with WT *Salmonella* (Fig 1D), confirming the functionality of Em-PAK1. As expected, ΔSopE1/E2 invasion was greatly reduced in all conditions, as SopE1/E2 are required to trigger the actin rearrangements that drive uptake.

To assess whether *Salmonella* activates PAK during infection, ΔPAK1 cells expressing Em-PAK1 were serum starved to reduce the starting level of phosphorylated (active) PAK, then either treated with control FBS, or infected for 20 minutes with WT or ΔSopE1/E2 *Salmonella*. Em-PAK1 was then extracted from cell lysates using a GFP Trap (Chromotek) and samples immunoblotted using a phospho-PAK1$^{Ser144}$ antibody. Serine 144 lies within the autoinhibitory domain of PAK, it is phosphorylated upon GTPase binding, and helps stabilise PAK in the open confirmation [7], thus is a good indicator of the levels of open active PAK in the cell. WT *Salmonella* caused a significant increase in the levels of phosphorylated PAK compared to the serum starved control, equivalent to that seen for FBS treatment, while the ΔSopE1/E2 strain did not (Fig 1E). Collectively these results show that PAK is involved in *Salmonella* uptake and is recruited and activated in a SopE1/E2-dependent manner.

## PAK Kinase domain is dispensable for *Salmonella* invasion

In an attempt to determine the role PAK plays during *Salmonella* invasion, various PAK1 constructs were transiently overexpressed in ΔPAK1 cells (S2A Fig), which were then infected with WT *Salmonella* (Fig 2A). The introduction of either WT or constitutively active PAK1 (PAK1$^{L107F}$, which is not autoinhibited and is instead constitutively "open") fully restored invasion, confirming that it was the loss of PAK1 causing the decrease in invasion. PAK1$^{H83/86L}$, which is unable to bind GTPases, was incapable of restoring invasion, however, remarkably, the expression of either a kinase-dead PAK1 construct (PAK1$^{K299R}$), or a constitutively open, kinase-dead variant (PAK1$^{K299R, L107F}$), completely restored *Salmonella* entry, suggesting that kinase activity is not required. This result was surprising as chemical inhibition of PAK kinase activity blocks *Salmonella* entry (Fig 1A). As these inhibitors target all Class I PAKs, it may

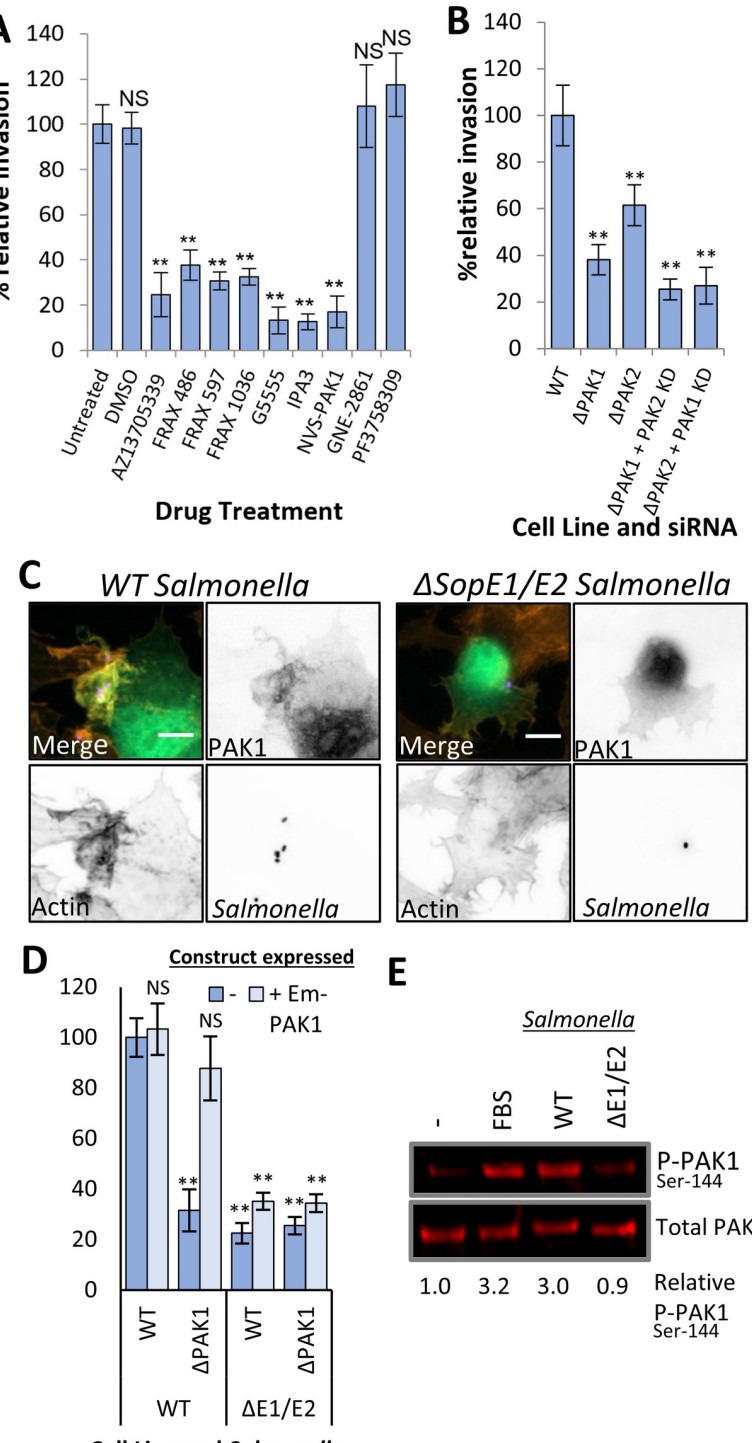

**Fig 1. Class I PAKs are crucial for *Salmonella* uptake. (A)** WT Hap1 cells were untreated, pretreated with DMSO (controls) or one of the PAK inhibitors AZ13705339, FRAX 486, FRAX 597, FRAX 1036, G5555, IPA3, NVS-PAK1, GNE-2861 or PF3758309 prior to infection with WT *Salmonella* for 20 mins. Invasion values are relative to those in untreated Hap1 cells. **(B)** Invasion of *Salmonella* (20 mins) into WT, ΔPAK1, ΔPAK2 Hap1 cells as well as ΔPAK1 cells pretreated with PAK2 siRNA and ΔPAK2 cells pretreated with PAK1 siRNA. All values relative to invasion into WT Hap1 control cells. **(C)** Fluorescence microscopy of ΔPAK1 Hap1 cells expressing Emerald-PAK1 (green), infected with either WT or ΔSopE1/E2 *Salmonella* (both 10 mins) that had been prestained with Alexa-Fluor 350 (blue). Cells also stained with Texas-Red Phalloidin to visualise actin (red). Scale bar is 10 μm. **(D)** Invasion of WT and ΔSopE1/E2

*Salmonella* (20 mins) into WT and ΔPAK1 Hap1 cells, either mock-transfected (-) or transfected with Emerald-tagged PAK1 (+ Em-PAK1). Invasion values relative to WT *Salmonella* invasion into control WT Hap1 cells. **(E)** Immunoblot of total (Total PAK) and active (P-PAK1^Ser144^) Emerald-PAK, isolated from transfected ΔPAK1 Hap1 cells using a GFP-Trap. Cells were serum starved overnight, then either treated with FBS, or infected with WT or ΔSopE1/E2 *Salmonella* (20 mins), as indicated. The level of active PAK relative to total PAK, calculated from band intensities (relative P-PAK1^Ser144^), is indicated. All Error bars indicate SD. NS–no significant difference, ** —P <0.01, * P <0.05 (ANOVA followed by a post hoc Dunnett's comparison).

therefore be necessary for there to be some PAK kinase activity in the cell, even if the PAK1 recruited by *Salmonella* does not have any kinase activity of its own. This activity is likely required for the phosphorylation of residues such as Serine 144 that maintains the open active

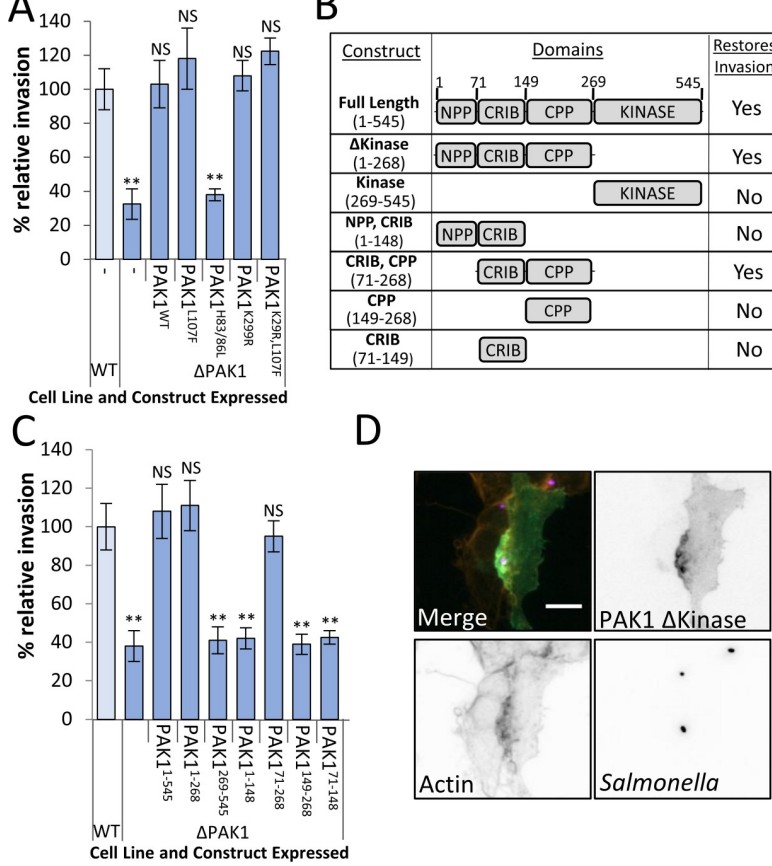

**Fig 2. PAK kinase domain is dispensable for *Salmonella* uptake. (A)** Invasion of WT *Salmonella* (20 mins) into WT and ΔPAK1 cells, or ΔPAK1 cells expressing either WT PAK1 (PAK1^WT^), constitutively open PAK1 (PAK1^L107F^), PAK1 that cannot bind GTPases (PAK1^H83/86L^), kinase dead PAK1 (PAK1^K299R^), kinase dead and constitutively open PAK1 (PAK1^K299R,L107F^). Invasion values relative to invasion in WT control cells. **(B)** Table summarising domain structure of PAK1: An N-terminal polyproline domain (NPP), a GTPase binding/autoinhibitory domain (CRIB/AID), a central polyproline domain (CPP) and a kinase domain (kinase). Also shown is the effect these constructs have on invasion when expressed in ΔPAK1 Hap1 cells, from the data in **(C)**. **(C)** Invasion of WT *Salmonella* (20 mins) into WT and ΔPAK1 Hap1 cells, as well as ΔPAK1 cells expressing PAK1 truncations summarised in (B): Full length (PAK1^1-545^), Δkinase (PAK1^1-268^), Kinase (PAK1^269-545^), NPP,CRIB/AID(PAK1^1-148^), CRIB/AID, CPP (PAK1^71-268^) CPP (PAK1^149-268^) and CRIB/AID (PAK1^71-148^). Invasion values relative to WT control cells. **(D)** Fluorescence microscopy of ΔPAK1 Hap1 cells transfected with Emerald-tagged PAK1Δkinase (green), infected (10 mins) with Alexa Fluor 350-stained *Salmonella* (blue). Actin stained with Texas-Red Phalloidin (red). Scale bar is 10 μM. All Error bars indicate SD. NS–no significant difference, ** —P <0.01, * P <0.05 (ANOVA followed by a post hoc Dunnett's comparison).

conformation of PAK1. Consistent with this, both WT PAK1 and PAK1$^{K299R}$ are phosphorylated in response to *Salmonella* invasion, but neither are in G5555-treated cells (S2C Fig). It is therefore possible that although kinase activity does not play a direct role in the signalling downstream of PAK, it does play a role in adopting an invasion-permissive conformation.

In order to explore this possibility further, and determine which domains of PAK are important, various PAK1 truncations were transiently overexpressed in ΔPAK1 cells (Figs 2B and S2B) and *Salmonella* invasion assays were carried out as before (Fig 2C). Consistent with the findings above, PAK1 lacking its kinase domain (PAK1$^{1-268}$) was able to fully restore *Salmonella* invasion to levels observed in WT cells, whereas the kinase domain alone (PAK1$^{269-545}$) could not. A construct encompassing the N-terminal polyproline region (NPP) and CRIB/AID (PAK1$^{1-149}$), could not restore invasion, however the CRIB/AID and central polyproline region together (CPP; PAK1$^{71-268}$) was also able to fully restore invasion. Neither CRIB/AID nor CPP alone (PAK1$^{71-149}$ and PAK1$^{150-268}$ respectively) had any effect.

To examine the requirements for PAK recruitment to *Salmonella entry foci*, Emerald-tagged versions of the PAK1 truncations were expressed in ΔPAK1 Hap1 cells, and then infected with WT *Salmonella* for 10 minutes. PAK1$^{1-268}$ (lacking the kinase domain) was recruited to the sites of bacterial invasion equivalently to full-length PAK1 (Fig 2D), as were all forms of PAK1 containing the CRIB/AID (S2D Fig). Conversely, truncations of PAK1 without a CRIB/AID were not recruited, with the kinase domain and CPP alone appearing mostly cytosolic. Collectively these data show that the CRIB/AID is required to localise PAK1 to invasion foci, and that this domain along with the CPP is required to promote *Salmonella* entry, with kinase activity, and indeed the kinase domain, being dispensable.

## PAK kinase and central polyproline domain have opposing effects on *Salmonella* invasion

Consistent with the kinase domain being dispensable, the kinase-dead mutant (PAK1$^{K299R}$) not only restored invasion in ΔPAK1 cells but its over-expression also enhanced uptake into WT cells (Fig 3A). A kinase-active mutant (PAK1$^{T423E}$) could not restore invasion in ΔPAK1 cells and surprisingly actually inhibited invasion when expressed in WT cells. This suggests that the kinase activity of PAK is not just dispensable, but that it is inhibitory for *Salmonella* uptake. The expression of the kinase domain alone (Fig 3B) failed to restore invasion in ΔPAK1 cells and despite kinase activity being inhibitory, its expression in WT cells had no significant effect on invasion (Fig 3C). This may be because the kinase domain alone fails to be recruited by *Salmonella* (S2A Fig). To artificially localise it to the plasma membrane, the kinase domain was fused to either the CRIB/AID of PAK1, or as *Salmonella* is known to generate this lipid at sites of invasion, to the PI(3,4,5)P3-binding Pleckstrin homology (PH) domain of the Arf GEF ARNO (CRIB-kinase and PH-kinase respectively; Figs 3B and S3A). Both PH-kinase and CRIB-kinase failed to restore invasion in ΔPAK1 cells, and they also impeded invasion in WT cells by 55% and 35% respectively. The greater inhibitory activity of the CRIB-kinase construct could be due to it also blocking the binding of endogenous native PAK to Rac1 and Cdc42, i.e., acting like a dominant-negative. Indeed, expression of the CRIB alone (and not the PH alone) impeded invasion to a similar extent in WT cells. These data demonstrate that when localised to the bacterial entry site, the PAK1 kinase domain reduces *Salmonella* invasion significantly.

The CPP domain of PAK is needed for *Salmonella* entry (Fig 2C), however when expressed alone it failed to restore invasion in ΔPAK1 cells, and significantly inhibited invasion when expressed in WT cells (Fig 3C). This could suggest that when expressed in isolation it acts as a

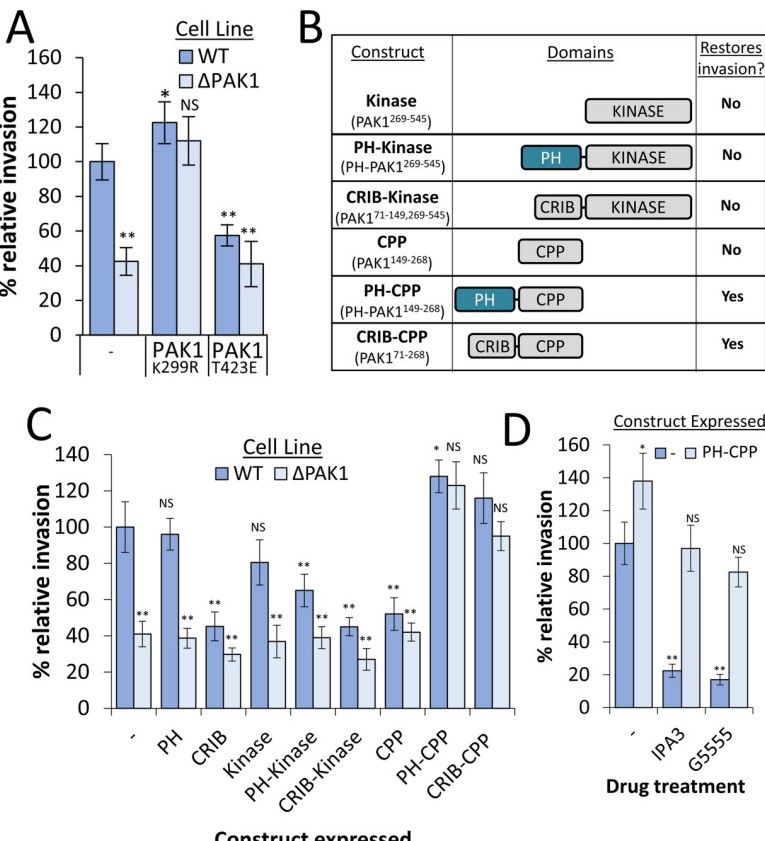

**Fig 3. PAK kinase and central polyproline domain have opposing effects on *Salmonella* invasion. (A)** Invasion of WT *Salmonella* (20 mins) into WT or ΔPAK1 Hap1 cells expressing the indicated construct: control (-), kinase dead PAK1 (PAK1^K299R) or constitutively active kinase PAK1 (PAK1^T423E). Invasion values relative to control WT Hap1 cells. **(B)** Table summarising PAK1 constructs and their effect on invasion in ΔPAK1 cells, from the data in **(C)**. PAK1 kinase domain (kinase), the kinase domain fused to a PH domain (PH-Kinase), CRIB/AID fused to kinase domain (CRIB-Kinase), central polyproline domain (CPP), CPP fused to a PH domain (PH-CPP) and CRIB/AID, CPP (CRIB-CPP). **(C)** WT *Salmonella* invasion (20 mins) into WT and ΔPAK1 Hap1 cells expressing the indicated construct, outlined in **(B)**. Invasion values relative to control WT Hap1 cells. **(D)** WT *Salmonella* invasion (20 mins) into control WT Hap1 cells (-) or the same expressing PH-CPP (+ PH-CPP). Cells were pretreated with either DMSO (control) or one of the PAK inhibitors IPA3 or G5555. Invasion values relative to control WT Hap1 cells. All Error bars indicate SD. NS–no significant difference, **—P <0.01, * P <0.05 (ANOVA followed by a post hoc Dunnett's comparison).

cytoplasmic sink for the proteins normally recruited by endogenous PAK. Like the kinase domain, the CPP alone does not localise at the membrane. As shown above (Fig 2D), when fused to the CRIB domain (CRIB-CPP), this domain fully restored invasion in ΔPAK1 cells, and drove a small increase in invasion in WT cells. Interestingly, the expression of CPP fused to the ARNO PH domain (PH-CPP) not only significantly promoted uptake in WT cells, but restored invasion to 123% of control in ΔPAK1 cells. PH-CPP could also restore *Salmonella* entry in WT Hap1 cells treated with the PAK inhibitors IPA3 or G5555 (Fig 3D), as well as in MEF and HeLa cells treated with these inhibitors (S3B and S3C Fig). When expressed in Hap1 cells Emerald-tagged PH-kinase, PH-CPP as well as CRIB-kinase were all successfully recruited by *Salmonella* after a 10-minute infection, confirming that an exogenous PH domain (and also the CRIB/AID of PAK1 itself) could successfully localise proteins to bacterial entry foci (S3D Fig). It seems therefore that when appropriately localised to the membrane, the PAK1 CPP alone is capable of driving *Salmonella* invasion.

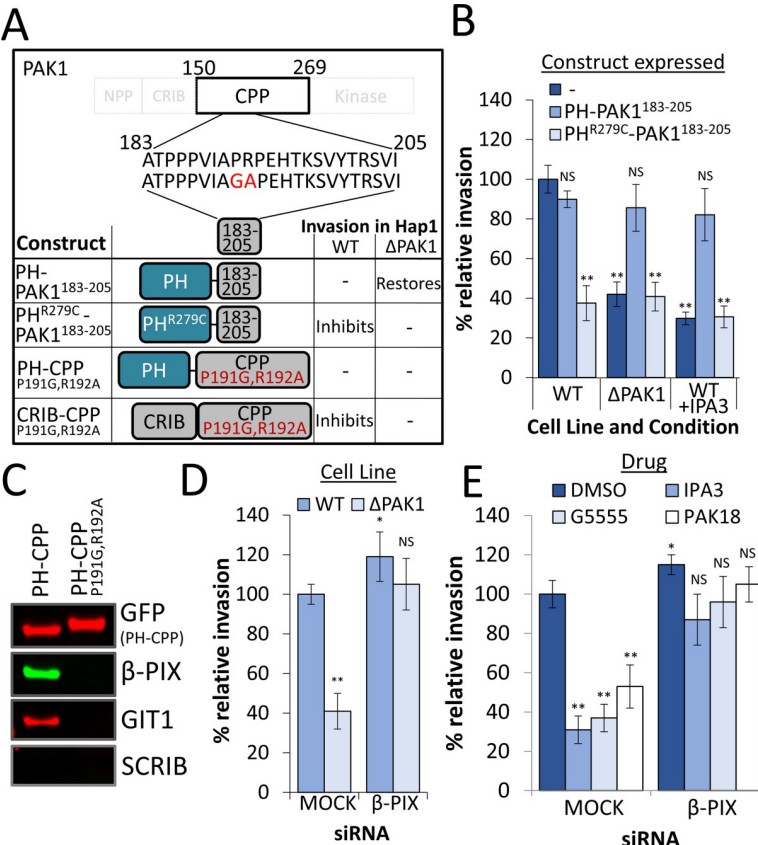

**Fig 4. The interaction between PAK and β-PIX is crucial for *Salmonella* invasion, yet β-PIX depletion restores invasion in the absence of PAK.** (A) The position and sequence of the central polyproline motif (PAK1[183-205]), and residues mutated within this region. Also shown are the constructs expressed in (B): PAK1 polyproline motif fused to a PH domain (PH- PAK1[183-205]), the same domain fused to a mutated PH domain incapable of interacting with phospho-lipids (PH[R279C]-PAK1[183-205]), CPP mutant unable to interact with SH3-containing proteins, fused to either PH domain (PH-CPP[P191G,R192A]), or the PAK1 CRIB domain (CRIB-CPP[P191G,R192A]). (B) Invasion of WT *Salmonella* into WT, ΔPAK1 or WT Hap1 cells pretreated with IPA3. Cells expressed the indicated construct. (C) Immunoblot of proteins that interact with PH-CPP and PH-CPP[P191G,P192A]. Emerald-tagged PH-CPP or PH-CPP[P191G,R192A] were expressed in ΔPAK1 Hap1 cells 48 hrs prior to being isolated using a GFP-Trap. Total extracted protein was detected using a GFP antibody, and β-PIX, GIT1 and SCRIB using corresponding antibodies. (D) Invasion of WT *Salmonella* into WT or ΔPAK1 Hap1 cells pretreated with mock or β-PIX siRNA (72 hrs prior to infection). Invasion is relative to mock siRNA-treated WT Hap1 cells. (E) Invasion of WT *Salmonella* into WT Hap1 cells pretreated with DMSO (control) or one of the PAK inhibitors IPA3, G5555 or PAK18, and also either pretreated with mock or β-PIX siRNA (72 hrs prior to infection). Invasion values are relative to mock siRNA, DMSO treated cells. All Error bars indicate SD. NS–no significant difference, ∗∗—P <0.01, ∗ P <0.05 (ANOVA followed by a post hoc Dunnett's comparison).

## The interaction between PAK and β-PIX is crucial during *Salmonella* invasion, yet β-PIX depletion restores invasion in the absence of PAK

Having identified the PAK CPP as crucial during *Salmonella* invasion we next sought to determine its function. The CPP contains a poly-proline motif (PAK1[183-205] –Fig 4A) which interacts with SH3 domains of other proteins. Expression of this motif fused to a PH domain for membrane localisation (PH-PAK1[183-205]) fully restored *Salmonella* uptake in ΔPAK1 cells (as well as in WT cells treated with IPA3) to the same level as in WT control cells (Figs 4B and S4B). The expression of the same construct with a mutated PH domain (rendering it unable to bind membranes—PH[R279C]-PAK1[183-205]) however, was unable to restore uptake. Interactions between this polyproline motif and SH3 domain-containing proteins can be disrupted with the mutation

P191G, R192A [16]. Expression of either PH-CPP$^{P191G,R192A}$ or CRIB-CPP$^{P191G,R192A}$ failed to restore uptake in both ΔPAK1 cells and IPA3-treated WT cells (S4A Fig). The expression of CRIB-CPP$^{P191G,R192A}$ also inhibited invasion in WT control cells, most likely due to it blocking Rac1 and Cdc42 from recruiting endogenous native PAK, as seen above for CRIB-containing constructs (Fig 3C). These data suggest that interactions between this poly-proline motif and SH3 domain-containing proteins at the plasma membrane are essential for *Salmonella* invasion.

To determine which proteins were recruited by this polyproline motif, the CPP domain was fused to the C2 domain of lactadherin (a phosphatidylserine-binding domain, henceforth "PSB"), which would allow us to load the purified protein onto lipid bilayers. Phosphatidylcholine:phoshatidyserine (PC:PS) coated silica microspheres [30] were loaded with either control purified His-PSB alone or His-PSB-CPP. Loaded microspheres were incubated in porcine brain extract for 20 minutes, before being isolated, washed, and recruited proteins analysed by mass spectrometry (See S1 Data). The only proteins that showed significant specific binding to CPP were β-PIX, two isoforms of the β-PIX binding partner GIT (GIT1 and GIT2), and SCRIB, a scaffold protein also known to form a tight complex with β-PIX [31]. To confirm these interactions in Hap1 cells, we expressed Emerald-tagged PH-CPP and PH-CPP$^{P191G,R912A}$, then isolated these proteins using a GFP trap. PH-CPP, but not the SH3-binding mutant PH-CPP$^{P191G,R912A}$, was able to efficiently recruit β-PIX as well as the β-PIX binding partner GIT1. Recruitment of SCRIB however was not detected (Fig 4D).

Consistent with the importance of the PAK-β-PIX interaction, treatment of cells with the inhibitor PAK18 (a peptide corresponding to PAK residues 187–203, which disrupts binding between PAK and β-PIX, impeded *Salmonella* invasion by 55%, with a negative control peptide (PAK18$^{-ve}$) having no effect (S4C Fig). Likewise, expression of the PAK1-binding β-PIX SH3 domain also impeded *Salmonella* invasion (by 75%), with a non-specific SH3 domain (Abi1-SH3) having no significant effect (S4D Fig)

It is clear that PAK must interact with β-PIX in order to drive *Salmonella* uptake, yet despite this, to our surprise, depletion of β-PIX using siRNA not only failed to impede *Salmonella* invasion, but actually promoted it by 20% (Fig 4D). Even more intriguingly, when β-PIX was depleted in ΔPAK1 cells *Salmonella* invasion was restored to levels comparable to those in WT cells. Control and β-PIX depleted cells were also treated with the PAK inhibitors IPA3, G5555 or PAK18, and whilst all inhibitors impeded invasion by at least 70% in control cells, in β-PIX depleted cells each inhibitor failed to significantly reduce invasion. The depletion of β-PIX also successfully restored invasion in MEF and HeLa cells treated with G5555 or IPA3 (S4E and S4F Fig). Similarly, in a β-PIX knockout cell line (Δ-βPIX) *Salmonella* invasion was enhanced by 35% relative to the control, and the three PAK inhibitors had no significant negative effect on invasion (S4G Fig). This finding that β-PIX depletion restores invasion in the absence of Class I PAKs, suggests that the interaction between PAK and β-PIX may be necessary to impede some negative effect β-PIX has on invasion.

## β-PIX GEF activity at focal adhesions negatively regulates *Salmonella* invasion

Consistent with the idea that β-PIX negatively regulates *Salmonella* uptake we found that its overexpression in WT Hap1 cells impeded invasion by over 50% (Fig 5B), and to a similar extent in both MEF and HeLa cells (S5A and S5B Fig). The isoform of β-PIX used here has five distinct domains, an N-terminal SH3 domain (which binds PAK's CPP) a Dbl-homolgy domain with Rac1 and Cdc42 GEF activity (DH), a lipid binding domain (PH), a GIT interaction domain (GID), and a C-terminal Region with a less well-defined function (CTR) [32]. To investigate which of these domains were responsible for inhibiting *Salmonella* invasion various

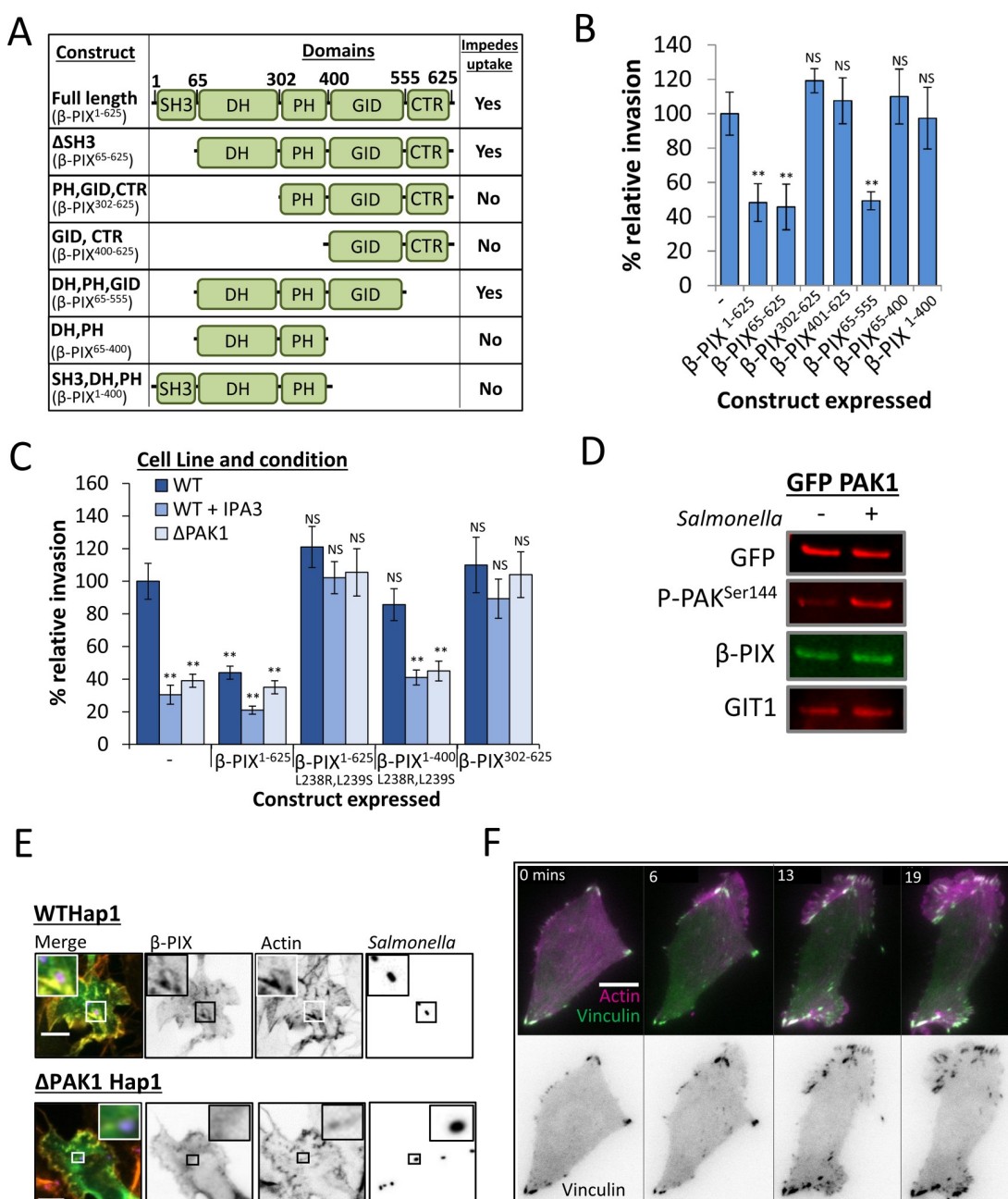

**Fig 5. β-PIX GEF activity at Focal adhesions negatively regulates *Salmonella* invasion. (A)** Schematic showing the major β-PIX domains, and the truncated variants of β-PIX used in **(B)**. β-PIX has 5 distinct domains: an N-terminal PAK-binding SH3 domain (SH3), a dbl-Homology GEF domain (DH), a lipid binding Pleckstrin-homology domain (PH), a GIT interaction domain (GID) and a C-terminal region (CTR). Table also summarises the effect the constructs have on invasion when expressed in WT cells, from the data in **(B)**. **(B)** Invasion of WT *Salmonella* into WT Hap1 cells expressing the indicated construct. Invasion values are relative to control WT cells. **(C)** Invasion of WT *Salmonella* into WT, ΔPAK1 and WT Hap1 cells pretreated with IPA3. Cells expressed the indicated construct: control (-), full length β-PIX (β-PIX$^{1-625}$), full length β-PIX with a mutation in the DH domain that ablates GEF activity (β-PIX$^{1-625\ L238R,L239S}$), the SH3, DH and PH domains of β-PIX with the same mutation (β-PIX$^{1-400\ L238R,L239S}$) or β-PIX lacking its SH3 or DH domain (β-PIX$^{302-625}$). Invasion values are relative to control WT Hap1 cells. **(D)** Immunoblot of proteins that interact with PAK1. Emerald-tagged PAK1 was expressed in ΔPAK1 Hap1 cells 48 hrs prior to being isolated using a GFP-Trap. Cells were uninfected or infected for 20 mins with WT *Salmonella*. Total extracted protein was detected using a GFP antibody, and β-PIX, GIT1 and P-PAK$^{ser144}$ using corresponding antibodies. **(E)** Fluorescence microscopy of WT and ΔPAK1 Hap1 cells expressing Emerald-tagged β-PIX (green). Cells were infected with WT *Salmonella* (10 mins) that had been pre-stained with Alexa fluor-350 (blue). Actin was stained with Texas-Red-phalloidin (red). Insets magnify the indicated area. Scale bar is 10 μm. **(F)** Fluorescence microscopy of cells expressing Emerald-vinculin (green) and Apple-actin

(purple). Cells were infected with WT *Salmonella*, and still images taken from a live movie at 0, 6, 13, and 19 minutes post infection. Scale bar is 10 μm. All Error bars indicate SD. NS–no significant difference, **—P <0.01, (ANOVA followed by a post hoc Dunnett's comparison).

truncations were generated (Fig 5A) and transiently expressed in both WT and Δβ-PIX cells (S5C Fig). The expression of constructs lacking either (or both) the SH3 and CTR domain still impeded uptake to a similar extent as full length β-PIX, in both WT (Fig 6B) and Δβ-PIX cells (S5D Fig). Conversely the expression of constructs that lacked either the DH or GID domain failed to impede *Salmonella* uptake (summarised in Fig 5A). Therefore, in order to negatively regulate *Salmonella* uptake β-PIX requires both the DH and GID domain.

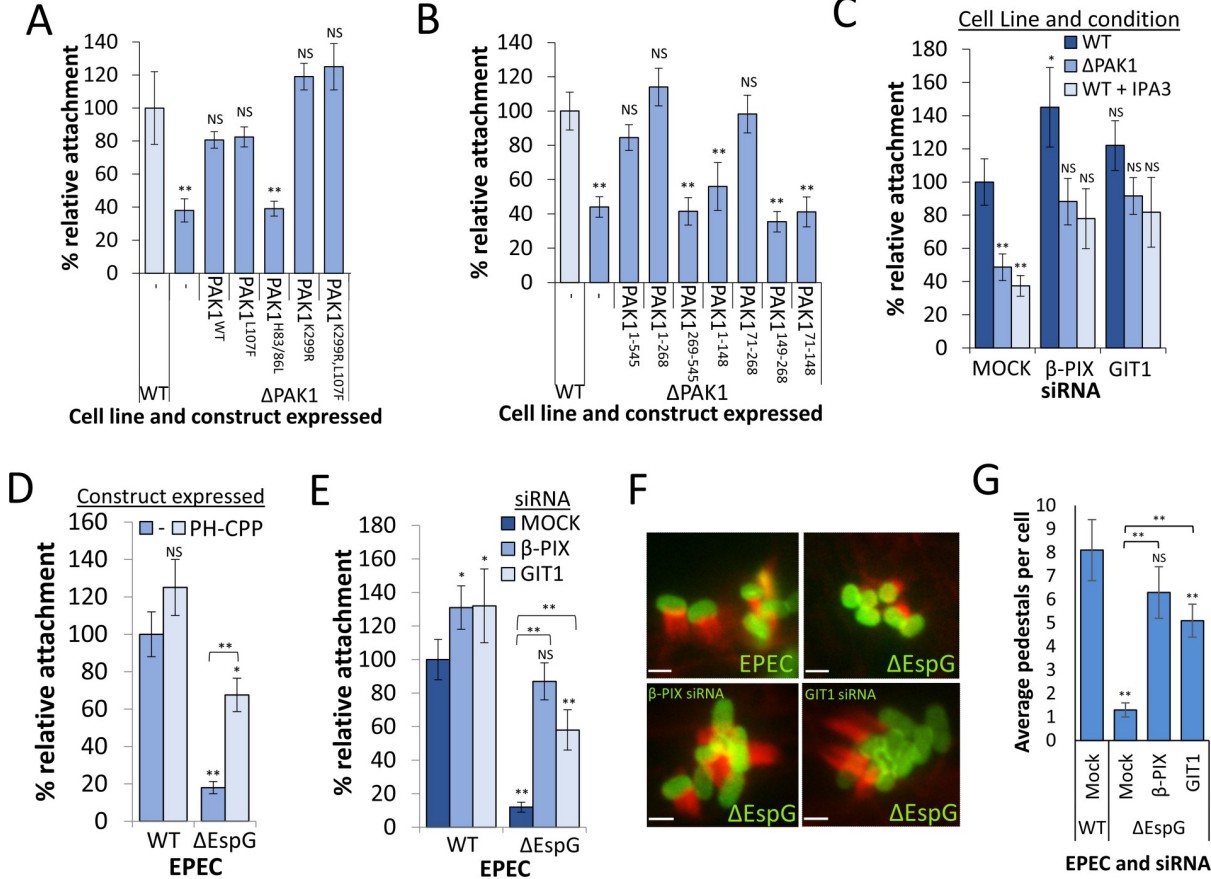

**Fig 6. PAK also drives EPEC attachment in a CPP dependent and kinase independent manner. (A)** Attachment of WT EPEC to ΔPAK1 Hap1 cells expressing the indicated PAK1 constructs, as described in (Fig 2A). Attachment values are relative to WT Hap1 cells. **(B)** Attachment of WT EPEC to ΔPAK1 Hap1 cells expressing the indicated PAK1 truncations, as described in (Fig 2C). Attachment values relative to WT Hap1 cells. **(C)** Attachment of WT EPEC to WT Hap1 cells, ΔPAK1 Hap1 cells or WT Hap1 cells pretreated with IPA3. All cells were pretreated with either mock, β-PIX or GIT1 siRNA 72 hrs prior to infection. Attachment values relative to WT Hap1 cells treated with mock siRNA. **(D)** Attachment of WT or ΔEspG EPEC to control WT Hap1 cells (-) or those expressing the CPP of PAK1 fused a PH domain (+PH-CPP). Attachment values relative to control WT Hap1. **(E)** Attachment of WT and ΔEspG EPEC to WT Hap1 cells pretreated with either Mock, β-PIX, or GIT1 siRNA 72 hrs prior to infection. Attachment values relative to attachment of WT EPEC to WT Hap1 cells treated with mock siRNA. **(F)** Fluorescence microscopy of pedestal formation by EPEC and ΔEspG EPEC (90 mins) on WT Hap1 pretreated with Mock, β-PIX or GIT1 siRNA 72hrs prior to infection. Actin stained with Texas Red phallodin (red), EPEC stained with an anti-intimin antibody (green). Scale Bar is 2 μm. **(G)** Average number of pedestals formed per cell by EPEC and ΔEspG EPEC (90 mins) on WT Hap1 cells pretreated with Mock, β-PIX or GIT1 siRNA 72hrs prior to infection. All Error bars indicate SD. NS–no significant difference, **—P <0.01, * P <0.05 (ANOVA followed by a post hoc Dunnett's comparison or post-hoc Tukey).

To determine whether the GEF activity of the DH domain was responsible for impeding *Salmonella* uptake we also constructed a GEF-dead mutant of β-PIX (β-PIX$^{L238R,L239S}$) [16]. The expression of β-PIX$^{L238R,L239S}$ not only failed to impede invasion in either WT or Δβ-PIX but actually promoted it by 22% in WT cells (S5E Fig), suggesting that the GEF-dead version can act as a dominant-negative for endogenous β-PIX. Consistent with this, expression of β-PIX$^{L238R,L239S}$ could fully restore *Salmonella* uptake to control levels in both ΔPAK1 and IPA3 treated cells (unlike WT full length β-PIX), as indeed could expression of β-PIX$^{(302–625)}$, a construct lacking the GEF domain (Fig 5C). However, a GEF-dead construct lacking the GID (β-PIX$^{1-400(L238R,L239S)}$) had no effect when expressed in these cells. Together this suggests that interaction with GIT is required for the β-PIX GEF activity to inhibit *Salmonella* entry, and for GEF-dead β-PIX derivatives to act as dominant-negative.

GIT, through an interaction with the protein Paxillin, is known to localise the β-PIX-GIT complex to Focal Adhesion (FA) complexes [33]. In agreement with this, Emerald-tagged full length β-PIX (Em-β-PIX) localised to adhesion complexes when expressed in Hap1 cells, as did all variants of β-PIX containing an intact GID (S5E Fig). In contrast, Em-β-PIX$^{1-400}$ (lacking a GID) appeared to be mostly cytosolic. We can therefore hypothesise that β-PIX exerts a negative effect on *Salmonella* invasion via its GEF activity when localised to FAs by GIT. To provide further evidence for this notion WT, ΔPAK1, and IPA3 treated WT Hap1 cells were pretreated with MOCK or GIT1 siRNA. *Salmonella* invasion was enhanced by 30% in GIT1-depleted WT cells, and this depletion also restored invasion to control levels in ΔPAK1 and IPA3-treated cells (S5F Fig). GIT depletion restoring invasion in cells lacking class I PAK's strongly suggests that when β-PIX interacts with GIT1 (and therefore localises to the FA), it inhibits *Salmonella* invasion. This inhibition can be overcome when PAK interacts with β-PIX, possibly through the recruitment of the β-PIX-GIT complex to the bacterial entry foci, away from the FA. To confirm that full length PAK interacts with β-PIX, and to determine whether this interaction changes during *Salmonella* invasion, ΔPAK1 cells expressing Em-PAK1 were infected with WT *Salmonella* for 20 mins. Em-PAK1 was isolated using a GFP-Trap, and the recruitment of β-PIX and GIT1 were assessed by immunoblotting. Whilst the levels of active PAK (P-PAK$^{ser144}$) increased relative to uninfected control cells, the levels of β-PIX and GIT bound to PAK did not noticeably change (Fig 5D). This would suggest that the mechanism that PAK employs to impede β-PIX may not rely on increased β-PIX-PAK binding, and instead could be due to a change in the localisation of the PAK-β-PIX complex.

β-PIX is known to modulate FA assembly [33,34], therefore its removal by PAK from the FA during *Salmonella* invasion may alter FA dynamics. Consistent with this, during *Salmonella* invasion Em-β-PIX is recruited to *Salmonella*-induced ruffles in WT cells, but not in ΔPAK1 cells (Fig 5E) and is recruited by *Salmonella* in MEF and HeLa cell (S5G Fig). Concomitant with this, a stark reorganisation of FAs occurs during the invasion process, with large Vinculin- and actin-rich FA complexes forming within the *Salmonella* induced ruffle (Fig 5F). To summarise, the data here demonstrates a novel pathway by which PAK, activated by SopE1/E2 during Salmonella invasion, recruits β-PIX to bacterial entry foci (away from FA's where its GEF activity is inhibitory) altering FA dynamics in order to drive the actin reorganisation required for efficient uptake of the bacteria.

## PAK also drives EPEC attachment in a CPP-dependent, and kinase-independent manner

As we have previously demonstrated the necessity of PAK in promoting the intimate attachment of EPEC to host cells [24], we sought to determine whether EPEC, like *Salmonella*, hijacks PAK in order to overcome the negative effect of β-PIX GEF activity at FAs. Almost

identically to *Salmonella* uptake, EPEC attachment is fully restored in ΔPAK1 cells when kinase-dead PAK (PAK1$^{K299R}$) is expressed (as it is when constitutively-open PAK—PAK1$^{L107F}$, constitutively-open kinase-dead PAK—PAK1$^{K299R,L107F}$, but not PAK that cannot bind GTPases—PAK1$^{H83/86L}$ is expressed—Fig 6A, representative image of attached EPEC–S6A Fig). Likewise, expression of either PAK1 lacking the kinase domain (PAK1$^{1-268}$), or a construct corresponding to the CRIB/AID and CPP domains (PAK1$^{71-268}$), also fully restored attachment in ΔPAK1 cells (Fig 6B). The expression of a kinase-active form of PAK (PAK1$^{T423E}$) impeded attachment by 40% in WT cells and failed to restore it in ΔPAK1 cells (S6A Fig). Finally, the expression of PH-CPP fully restored EPEC attachment in ΔPAK1 cells, with PH-kinase or CPP alone exerting a negative effect on attachment in WT cells (S6B Fig). Therefore, it appears that, as is the case for *Salmonella* invasion, PAK's role in EPEC attachment requires the CPP domain (correctly localised to the membrane), with the kinase domain not only being dispensable, but inhibitory. The ability to restore attachment by EPEC also translates into being able to form actin rich pedestals, with PAK1, PAK1$^{K299R}$ and PAK1ΔKinase restoring pedestals formation in ΔPAK1 Hap1 cells, and PAK1$^{T423E}$ failing to do so (S6C and S6D Fig).

The depletion of β-PIX using siRNA promoted the attachment of EPEC to WT cells by more than 40%, and the depletion of either β-PIX, or its binding partner GIT1 restored EPEC attachment in either ΔPAK1 cells or WT cells treated with the PAK inhibitor IPA3 to WT control levels (Fig 6C). This result indicates that PAK's role during EPEC attachment is to impede the negative effect of the β-PIX-GIT complex, as it is during *Salmonella* invasion.

EPEC utilises the effector protein EspG in order recruit activated PAK and promote intimate attachment to host cells [24,27]. In the absence of EspG attachment is very poor, therefore we sought to confirm that this is due to the inability to impede or alter the localisation of β-PIX via PAK recruitment. Whilst not fully restoring attachment to the level of WT EPEC, the expression of PH-CPP greatly promoted the attachment of ΔEspG EPEC to WT Cells (from just 19% of control attachment to 68%—Fig 6D). Depletion of β-PIX fully restored the ability of ΔEspG EPEC to attach to WT cells, to a level similar to WT EPEC. Depletion of the β-PIX binding partner GIT1 also promoted attachment of ΔEspG EPEC, albeit to a lesser extent (Fig 6E). Depletion of either β-PIX or GIT1 also greatly promoted ΔEspG pedestal formation as expected (Fig 6F and 6G). This data strongly implies that in order to promote attachment to host cells, EPEC delivers the effector EspG, which recruits PAK, likely resulting in the recruitment of β-PIX-GIT complexes away from FAs, where they are normally inhibitory to the process.

In conclusion it seems that both *Salmonella* and EPEC utilise PAK in the same kinase-independent manner to drive the actin reorganisation required for efficient infection of host cells.

## Discussion

Collectively the data presented here allows us to propose a model outlining how *Salmonella* hijacks PAK in order to promote uptake into host cells (Fig 7). *Salmonella* injects the effector proteins SopE1/E2 which activate the small GTPases Rac1 and Cdc42, resulting in the recruitment and activation of Class I PAKs at sites of bacterial entry. PAK could be recruited from the cytosol as well as directly out of nearby FAs (where it resides in resting cells). PAK, via its central polyproline region (CPP), is able to interact with the Rac1 GEF β-PIX, and either recruits the β-PIX-GIT complex out of the FA or modifies β-PIX directly within the FA. Either way, the interaction between PAK and β-PIX prevents β-PIX from impeding the uptake of *Salmonella* and creates a local cellular environment permissive to the actin reorganisation required. In the absence of PAK, β-PIX resident in FAs is able to exert a negative effect on

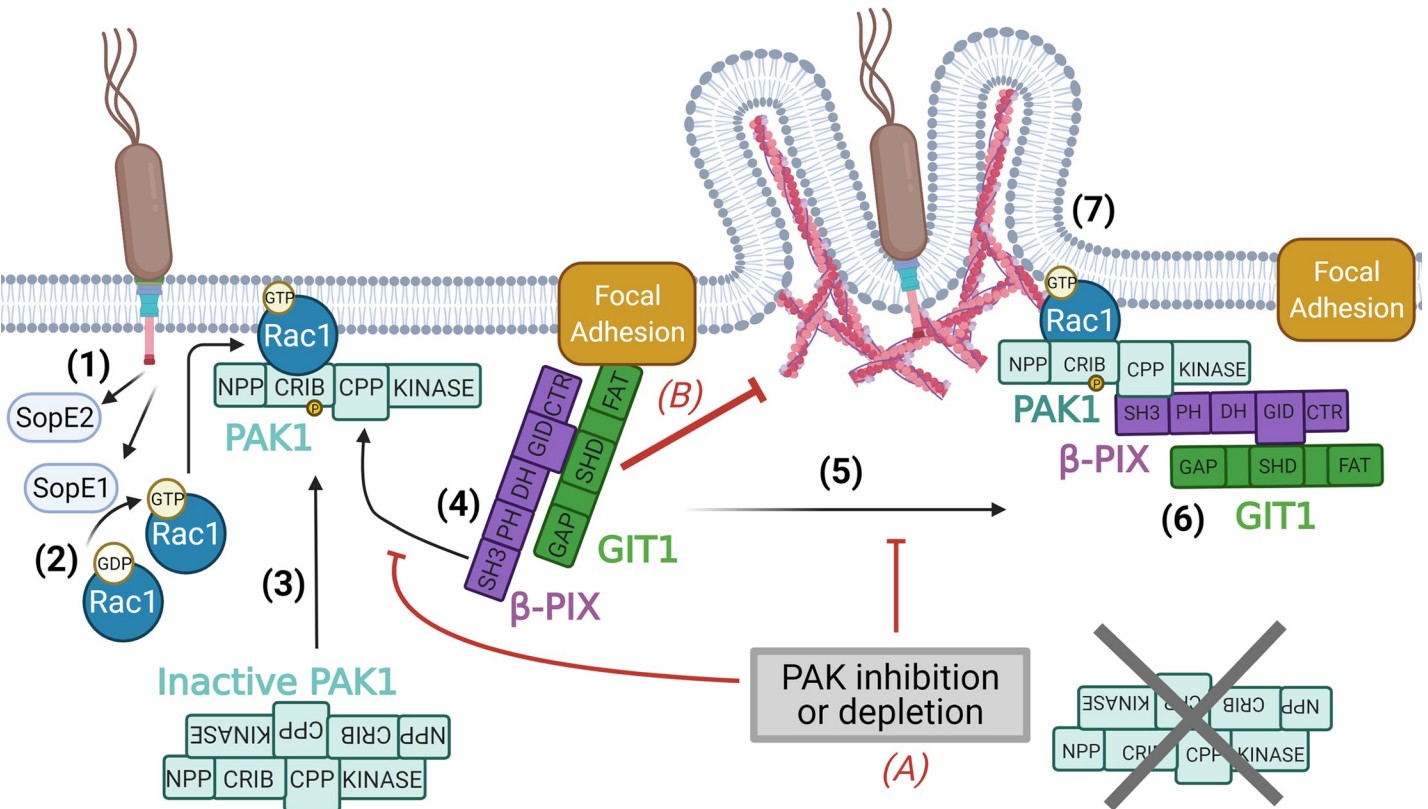

**Fig 7. Model: PAK drives *Salmonella* uptake.** (1) When in contact with the host cell *Salmonella* injects effector proteins into the cytoplasm using its T3SS. (2) Two of these effectors (SopE1 and SopE2) activate the small GTPases Rac1 and Cdc42. (3) Rac1 (or Cdc42) binds to the PAK CRIB domain, recruiting PAK to the plasma membrane. (4) PAK can recruit β-PIX through an interaction between its CPP and β-PIX's SH3 domain. (5) The β-PIX-GIT complex is pulled out of the FA. (6) Once removed from the FA β-PIX no longer exerts a negative effect on Salmonella uptake. (7) The local cellular environment is now conducive to the actin reorganisation that drives bacterial internalisation. (A) In the absence of PAK (through depletion or inhibition) (B) β-PIX remains in the FA where it impedes *Salmonella* invasion either via modulating FA dynamics, and/or by sequestering Rac1 away from the bacterial entry site. This inhibition is dependent on β-PIX GEF activity. Created with BioRender.com.

*Salmonella* uptake through its GEF activity. This is potentially caused by the mislocalisation of Rac1 away from invasion sites, as well as by the local manipulation of FA dynamics, both of which may impede the actin assembly necessary for *Salmonella* uptake.

Class I PAKs are serine/threonine kinases that are key regulators of the actin cytoskeleton, therefore it comes as no surprise that many pathogens subvert PAK to promote infection [23]. Its role in *Salmonella* uptake though does comes as a surprise since previous research, which outlined a key kinase-dependent role for PAK in the activation of c-Jun $NH_2$-terminal kinase (JNK) during infection, found no obvious role for PAK in the invasion process [20]. These conclusions relied on the expression of dominant-negative PAK mutants which, as the authors themselves concede, are often not wholly effective. Indeed, one of the derivatives reported to have no effect on uptake was a kinase-dead version, a result we confirm (Fig 2A) and that is actually consistent with PAK's kinase activity being dispensable. Their research does however confirm that PAK is activated in a type three secretion system (T3SS) dependent manner during the first 30 minutes of infection, and that the expression of a variant of Cdc42 that cannot bind PAK does impede uptake, consistent with our findings here. More recent research has demonstrated that PAK1 activated during *Salmonella* infection plays a vital role in the stimulation of NF-κB, in order to establish pro-inflammatory signalling after cell invasion [21]. Throughout our study, we exploit a range of drugs, knockout cell lines and siRNA experiments

to clearly establish a role for PAK in the entry of *Salmonella* into non-phagocytic cells, we confirm many of our key results in HeLa and MEF cells also.

We demonstrate that the chemical inhibition of Class I but not Class II PAKs greatly impedes the invasion of *Salmonella*, something we previously found to be the case for EPEC and EHEC attachment [24]. We also outline a more prominent role for PAK1 over PAK2 through the use of knockout cells. Surprisingly, despite the kinase activity of PAK being previously shown to be an important regulator of the actin cytoskeleton [1]- as well as specifically of macropinocytosis [22], an event that shares many features with *Salmonella* invasion, our results clearly demonstrate that the role of PAK during *Salmonella* entry is independent of kinase activity, and instead requires only the CRIB and CPP domains. We show that chemical inhibitors of PAK that interfere with kinase activity actually impede *Salmonella* invasion, which appears at odds with the finding that the kinase domain of PAK is dispensable for uptake. These kinase inhibitors however target all Class I PAK kinase activity in the cell, and therefore prevent phosphorylation of residues such as Ser144 that leads to PAK becoming open and active. We would expect that these inhibitors block all PAK functions, and not just those associated with kinase activity. We therefore propose that whilst some PAK kinase activity is required for the initial opening and activation of PAK1, once recruited by *Salmonella* (or EPEC), the kinase domain plays no role in the downstream signalling required to drive uptake or attachment.

Although surprising, the kinase-independent role of PAK is not unprecedented. PAK has previously been proposed to function as a scaffold [16], mediating the recruitment of several effectors to sites of Rac1 activation. The expression of kinase-dead PAK can also induce lamellipodia formation [35–37], and in fact this mutant of PAK may be even more efficient at driving the generation of Rac1 dependent ruffles than wildtype [33]. Further to being dispensable for efficient uptake of *Salmonella*, we have shown that the kinase domain of PAK may actually inhibit invasion, partially opposing the kinase-independent pro-invasive function. Several PAK kinase substrates inhibit actin turnover such LIM-Kinase (LIMK), (which phosphorylates cofilin) [14], and MLC [38], which may explain why the kinase activity inhibits *Salmonella* invasion. Further work is required to define the mechanisms that control the balance between the potential stimulatory and inhibitory pathways that exist downstream of PAK.

We demonstrate that the CPP of PAK (when appropriately localised) is the sole requirement for PAK to promote efficient *Salmonella* uptake. It is well established that the CPP of PAK interacts with the Rac1/Cdc42 GEF β-PIX[16], and we too demonstrate this to be the case (Figs 4C and 5D). Initially we assumed that the GEF activity of β-PIX, when recruited to entry sites, could activate additional Rac/Cdc42, promoting invasion in a manner analogous to the role previously reported for the bipartite Rac1 GEF ELMO1/DOCK180 [39]. In fact, we demonstrate that β-PIX GEF activity impedes *Salmonella* uptake (Figs 5B, 5C, S5A, S5B and S5D). It could be that β-PIX activates Rac1 at the wrong location to promote uptake. Consistent with this, β-PIX GEF activity only inhibited uptake when β-PIX was able to interact with its well characterised binding partner, GIT1 [33]. As this interaction leads to localisation at FAs, it is therefore possible that β-PIX diverts large amounts of Rac1 away from the *Salmonella* entry foci. β-PIX has indeed been shown to localise Rac1 to FAs [40], and previous research has reported that in cells lacking Paxillin, a protein that links β-PIX-GIT to the FA [41], *Salmonella* uptake is enhanced [42], consistent with our findings that uptake is increased in β-PIX and GIT1 depleted cells. *Salmonella*-activated PAK, perhaps via competition for Rac1 binding by β-PIX [43], or by recruiting β-PIX (and indeed GIT1) out of the FA, may overcome this Rac1 mislocalisation. Whilst we demonstrate as expected that Rac1 is recruited to sites of *Salmonella* entry, determining whether PAK inhibition prevents this from occurring is difficult, since PAK inhibited cells in general do not form actin rich ruffles (S7A Fig). This aside, it is

still feasible that Rac1 mislocalisation caused by β-PIX is something that PAK is able to overcome during infection.

An alternative explanation is that by removing β-PIX from the FA, *Salmonella* may alter FA dynamics in a way conducive with the actin rearrangements required for uptake. Depletion of β-PIX [44,45] or overexpression of mutant variants lacking GEF activity [46,47] (both of which we have shown enhance *Salmonella* uptake) have been demonstrated to alter FA dynamics, and we show that FAs are remodelled during *Salmonella* invasion (Fig 5F), concomitant with β-PIX recruitment to entry foci (which we show occurs only in cells that express PAK1).

Consistent with the idea that FA dynamics are important, the FA components focal adhesion kinase (FAK), and p130Cas have previously been implicated in *Salmonella* uptake, with invasion being greatly impaired in FAK knockout cells [42]. Interestingly like PAK, the kinase activity of FAK was not required to restore *Salmonella* invasion, and we show that impeding FAK's kinase activity through the use of a chemical inhibitor [48] (S7B Fig), or through the expression of FAK-related nonkinase (FRNK [49]–S7C Fig), restores invasion in ΔPAK1 and IPA3 treated cells. This reveals a possible link between altering FA dynamics and promoting *Salmonella* uptake. However, we could not see any obvious difference in the accumulation of the FA component vinculin in the absence of PAK1 or β-PIX (S7D Fig). Whether *Salmonella* directly alters adhesions in and around the invasion foci or indirectly via changes to the state of the actin cytoskeleton remains to be determined. It is of course possible that the effects seen on *Salmonella* entry are due to combination of FA alterations and potential changes in Rac1 availability outlined above.

*Salmonella* and EPEC both manipulate the actin cytoskeleton to drive very distinct outcomes, yet fascinatingly they both require kinase-independent functions of PAK to achieve this. This suggest that what PAK does may not directly affect the kind of actin structures generated. Instead, PAK seems to provide a suitable environment in which other effector proteins (such as SopB, SipA and SopE in *Salmonella*, and Tir in EPEC) can carry out actin assembly to achieve the desired result. We propose that the modification of FA dynamics, (and/or the change in Rac1 localisation) achieved through the recruitment of the β-PIX-GIT complex out of the FA by PAK's CPP, allows *Salmonella* and EPEC to reorganise the actin cytoskeleton to achieve uptake or intimate attachment respectively. This novel kinase-independent mechanism for PAK may not just be exclusive to the two pathogens we have investigated, with other bacteria and viruses potentially hijacking PAK to promote infection. It remains to be seen whether this pathway is also involved in normal cellular regulation of FA dynamics, or indeed more intriguingly in aberrant conditions, such as those in cancerous cells where PAK is often overexpressed or abnormally activated [50,51]. PAK's are important pharmacological targets, therefore our discovery of a novel kinase-independent pathway may be of considerable interest for researchers when designing new drugs.

It is certainly clear that a greater understanding of how PAK modifies both the microenvironment, and FA dynamics is required. What is also certain though, is that the relationship between PAK and β-PIX may be more complicated than previously believed, and at least in terms of pathogen manipulation of the actin cytoskeleton, be far more important than previously appreciated.

## Methods

### Bacterial strains

WT *Salmonella* (*Salmonella enterica* serovar Typhimurium, SL1344) was a kind gift from Jean Guard-Petter, the isogenic ΔSopE1ΔSopE2 was a kind gift from Wolf-Dietrich-Hardt. WT

EPEC (E2348/69) and the isogenic mutant ΔespG1/ΔespG2 (ΔEspG) were kind gifts from Feng Shao.

## Plasmids

All Emerald tagged constructs were generated using the Gateway system (Invitrogen). Emerald PAK1, PAK1[L107F], PAK1[H83/86L] were generated from variants previously described [24]. Em-Tagged PAK1[K299R] and PAK1[T423E] were generated by subcloning from pCMV6M-PAK1 K299R (Addgene plasmid # 12210; http://n2t.net/addgene:12210; RRID:Addgene_12210) and pCMV6M-PAK1 T423E (Addgene plasmid # 12208; http://n2t.net/addgene:12208; RRID: Addgene_12208) which were a kind gift from Jonathan Chernoff. PAK1[K299R L107F] was generated using site-directed mutagenesis (SDM) with PAK1[K299R] as a template. All PAK1 truncations were subcloned from Em-PAK1 as a template. Emerald-tagged-β-PIX was generated by subcloning from Flag-betaPIXa, which was a kind gift from Rich Horwitz (Addgene plasmid # 15234; http://n2t.net/addgene:15234; RRID:Addgene_15234). All β-PIX truncations were subcloned from Em-β-PIX. Em-β-PIX[L238R,L239S] was generated using SDM with Em-β-PIX as a template. DNA encoding hybrid proteins CRIB-Kinase, PH-Kinase, PH-CPP, PH-PAK1[183-205], PSB-CPP, PSB and FRNK were synthesized (IDT Technologies) and subcloned into expression vectors using the Gateway system. PH[R279C]-PAK1[183-205], CRIB-CPP[P191G,R912A] and PH-CPP[P191G,R192A] were generated using SDM with Em-PH-PAK1[183-205], Em-CRIB-CPP and Em-PH-CPP as templates respectively. Emerald-Vinculin (Addgene plasmid # 54303; http://n2t.net/addgene:54303; RRID:Addgene_54303), and Apple-Actin (Addgene plasmid # 54862; http://n2t.net/addgene:54862; RRID:Addgene_54862) were gifts from Michael Davidson.

## Antibodies

The antibodies used were supplied by the following: Chromotek (GFP, catalog no. 3h9), Abcam (P-PAK1,2,3[Ser144/141], ab40795; Tubulin, ab7291), Cell signalling Technology (PAK1, 2602; PAK2, 2608; GIT1, 2919; SCRIB, 4475) Santa Cruz Biotechnology (β-PIX, sc-393184). Rabbit anti-intimin was raised against full-length recombinant intimin by Diagnostics Scotland.

## Mammalian cell culture

WT Hap1 cells and the sequence-verified knockouts ΔPAK1 (HZGHC000160c012), ΔPAK2 (HZGHC000053c001) and Δβ-PIX (HZGHC8874) were all purchased from Horizon Discovery. They were maintained in Iscoves's modified Dulbecco's medium (IMDM) supplemented with 10% Fetal Bovine Serum (FBS) and 100 U/ml penicillin-streptomycin. HeLa cells and MEFs were maintained in Dulbecco's modified Eagle's medium (DMEM) with 10% FBS and 100 U/ml penicillin-streptomycin. Where specified, cells were pre-treated with chemical inhibitors 30 minutes prior to an experiment being carried out, these were: 50 μM IPA-3, 5 μM PAK-18, PAK-18 negative control (Sigma), 10 μM G5555, NVS-PAK1-1, AZ13705339, FRAX 486, FRAX 597, FRAX1036, GNE-2861, PF3758309, PF573228 (all Tocris). Where mentioned, cells were transfected using the Neon System (Invitrogen) following the manufacturer's instructions. All constructs were transiently expressed for 18 hrs prior to being used in experiments unless otherwise stated, approximately 3 μg of plasmid was used per transfection.

Small interfering RNAs (siRNAs) used to knockdown expression of specific genes were purchased from Qiagen. These were PAK1 (HS_PAK1_1, AAGATTAACTTGGATCTTCTA; HS_PAK1_2, ACCCTAAACCATGGTTCTAAA), PAK2 (HS_PAK2_1, ACGAGTAATTGT GAAGCATAA; Hs_PAK2_4, TGCAGTAGTATAAATCATGAA), β-PIX (Human) (HS_Arhgef7_5, AACAATCAACTGGTAGTAAGA; HS_Arhgef7_6, CAAGCGCAAACCTGAACG

GAA), β-PIX (Mouse) (Mm_Arhgef7_1, CTCCTTGGTGACACTAAATAA; Mm_Arhgef7_2, CAGCAAACACTTCATATTTAA) and GIT1 (HS_GIT1_5, CAGCCTTGACTTATCCGAA TT; HS_GIT1_6, GCCGCTGAGGATGTCCCGAAA). siRNAs were transfected using Oligo-fectamine (Life-Technologies) following the manufacturer's instructions, a final concentration of 10 nM was used for each pair of siRNAs, and cells were treated 72 hrs prior to experiments being carried out. (Confirmation of knockouts and knockdown efficiencies can be found in S1D and S4H Figs)

### *Salmonella* infection assays

Invasion was quantified using *Salmonella* carrying the pM975 plasmid (kind gift from Wolf-Dietrich-Hardt). Once internalised these bacteria express green fluorescent protein (GFP) via the SPI2 promoter, allowing for detection of internalised bacteria only. Unless stated otherwise, cells were infected for 20 minutes using *Salmonella* at a multiplicity of infection (MOI) of 100, washed twice in Phosphate Buffered Saline (PBS), then incubated in Gentamycin (100 μg/ml) containing media for 90 minutes (to kill extracellular bacteria–a gentamycin protection assay). Cells were subsequently fixed, stained with Texas-Red Phalloidin (Life-Tech) and DAPI, then visualised using immunofluorescence (IF) microscopy. The number of GFP-positive (internalised) bacteria were counted per cell (minimum 500 cells per condition). All experiments were carried out at least three times, geometric means calculated, and significance determined, a P of $<0.05$ (as determined by ANOVA followed by a post hoc Dunnett's comparison) was deemed significant. When analysing recruitment of Emerald-tagged constructs, *Salmonella* were stained using Alexa-Fluor-350 carboxylic acid succinimidyl ester (Lifetech, 15 minutes, 25°C), then washed in Tris (pH 7.4)-buffered saline before being used in a 10-minute infection (unless stated otherwise). Images were taken using wide-field IF, or total internal reflection (TIRF) microscopy as stated in the text.

### Fluorescence microscopy

TIRF microscopy was performed on an Olympus IX83 inverted microscope equipped with a Uapon 100x/ 1.49NA objective and Olympus IX3-SSU automated xy stage and IX3 Z-Drift Compensator. Images were taken using CellSens software with a 100 nm penetration depth and recorded on a Hamamatsu imageEM x2 EM-CCD camera. Cells were imaged in Leibovitz L15 medium without phenol red (ThermoFisher) and maintained at 37°C in a stage top incubator (Cell vivo, Pecon). Images for invasion, attachment and pedestal formations assays as well as Emerald tagged PAK and β-PIX recruitment by *Salmonella* were captured on an Olympus IX81 inverted microscope.

### EPEC infection assays

For adhesion assays cells were infected with WT or ΔEspG EPEC for 90 minutes (MOI ~ 50), cells were then washed twice in PBS, twice briefly in ice cold 200 mM glycine (pH 2), then again in PBS twice. Cells were fixed, stained with Texas-Red Phalloidin and DAPI, and adherent EPEC stained using an anti-intimin antibody. The number of adherent bacteria per cell were then counted using fluorescence microscopy (of at least 500 cells). To assess pedestal formation the same assay was carried out without the glycine wash step and the number of actin rich pedestals per cell were counted. All experiments were carried out at least three times, geometric means calculated, and significance determined by ANOVA followed by a post hoc Dunnett's comparison or post-hoc Tukey where appropriate, with a P of $<0.05$ considered significant.

## GFP trap

Constructs expressing Emerald-tagged proteins were transfected into Hap1 cells 48 hours prior to performing the GFP trap. The GFP trap was carried out using the manufacturer's instructions (Chromotek), resulting trapped proteins run on an SDS-PAGE gel before being transferred to PVDF membranes, and probed with appropriate antibodies. All immunoblots were visualised using a LI-COR Odyssey Fc imaging system, with band intensities determined using LI-COR Image studio software.

## In-vitro pulldowns on lipid coated beads

His-tagged PSB and PSB-CPP were expressed in *E. coli* Rosetta (Novagen) at 18˚C before affinity purification [52]. Pulldowns using lipid coated silica micropsheres were subsequently carried out as described [30]. Briefly, silica microspheres (Bangs Laboratories) were coated in a lipid bilayer made up of phosphatidylcholine (PC) and phosphatidylserine (PS) at an 80:20 molar ratio (Avanti Polar Lipids). PSB or PS-CPP were anchored to these membranes and incubated in a cell-free porcine brain extract for 15 minutes, before being washed extensively. Recruited proteins were identified by electrospray ionization liquid chromatography mass spectrometry (MS) (Mass Spectrometry Service, Cambridge Centre for Proteomics, University of Cambridge, Cambridge, UK).

## Supporting information

**S1 Fig. (A)** Summary of drugs used and their mechanisms of action. Invasion of WT *Salmonella* (20 mins) into **(B)** WT MEF cells or **(C)** WT HeLa cells, each either pretreated with DMSO (control) or one of the PAK inhibitors AZ13705339, FRAX 486, FRAX 597, FRAX 1036, G5555, IPA3, NVS-PAK1, GNE-2861 or PF3758309. Invasion values are relative to those in control DMSO treated MEFs or HeLas respectively. **(D)** Immunoblot of cell lysates depicting relative amounts of PAK1 and PAK2 in WT, ΔPAK1, ΔPAK2 as well as ΔPAK1 cells pretreated with PAK1 siRNA, and ΔPAK2 cells pretreated with PAK2 siRNA. Levels of PAK1 and PAK2, quantified from band intensities and normalised to total tubulin levels, are displayed at the bottom. **(E)** Fluorescence microscopy of MEF and HeLa cells expressing Emerald-PAK1 (green), infected with WT *Salmonella* (10 mins) that had been pre-stained with Alexa-Fluor 350 (blue). Cells also stained with Texas-Red Phalloidin to visualise actin (purple). Scale bar is 10 μm. All Error bars indicate SD. NS–no significant difference, $^{**}$—P <0.01, $^{*}$ P <0.05 (ANOVA followed by a post hoc Dunnett's comparison).
(TIF)

**S2 Fig.** Expression of Full length **(A)** or truncated **(B)** PAK constructs confirmed by immunoblotting of corresponding cell lysates using a GFP antibody (Red), and a tubulin antibody (Green) as a loading control. Endogenous detected using a PAK1 antibody. **(C)** Immunoblot of total (Total PAK) and active (P-PAK1$^{Ser144}$) Emerald-PAK1 or Emerald PAK1$^{K299R}$, isolated from transfected ΔPAK1 Hap1 cells using a GFP-Trap. Cells were serum starved overnight, then either left uninfected, or infected with WT *Salmonella* (20 mins) in the absence or presence of G5555. **(D)** Fluorescence microscopy of ΔPAK1 Hap1 cells expressing emerald-tagged PAK constructs indicated (green), infected (10 mins) with Alexa Fluor 350-stained WT *Salmonella* (blue). Actin stained with Texas-Red Phalloidin (red). Scale bar is 10 μm.
(TIF)

**S3 Fig. (A)** Expression of constructs (PH, PH-kinase, CRIB-kinase and PH-CPP) confirmed by immunoblotting of corresponding cell lysates using a GFP antibody (Red), and a Tubulin antibody (Green) as a loading control. Invasion of WT *Salmonella* (20 mins) into **(B)** WT

MEF cells or **(C)** WT HeLa cells, each either pretreated with DMSO (control), IPA3 or G5555, cells expressed either no construct (-) or PH-CPP. Invasion values are relative to those in control DMSO treated MEFs or HeLas respectively **(D)** Fluorescence microscopy of ΔPAK1 Hap1 cells expressing the Emerald-tagged PAK1 constructs indicated (green), infected (10 mins) with Alexa Fluor 350-stained WT *Salmonella* (blue). Actin stained with Texas-Red Phalloidin (red). Scale bar is 10 μm. All Error bars indicate SD. NS–no significant difference, **—P <0.01, * P <0.05 (ANOVA followed by a post hoc Dunnett's comparison). (TIF)

**S4 Fig. (A)** Invasion of WT *Salmonella* into WT, ΔPAK1 or WT Hap1 cells pretreated with IPA3. Cells expressed the indicated construct, as described in (Fig 4A) **(B)** Expression of constructs indicated confirmed by immunoblotting of corresponding cell lysates using a GFP antibody (Red), and a Tubulin antibody (Green) as a loading control. **(C)** Invasion of WT *Salmonella* into WT Hap1 cells treated with control DMSO (-), PAK inhibitor PAK18 or the negative control version of PAK18 (PAK18 -ve). Invasion values are relative to those in control DMSO-treated cells. **(D)** Invasion of WT *Salmonella* into WT Hap1 cells expressing the indicated construct: control (-), the SH3 domain of β-PIX (β-PIX-SH3), or the SH3 domain of Abi1 (Abi1-SH3). Invasion values are relative to those in control cells. Invasion of WT *Salmonella* (20 mins) into MEF **(E)** or HeLa **(F)** cells, pretreated with either DMSO, IPA3 or G5555, and also either pretreated with mock or β-PIX siRNA (72 hrs prior to infection). Invasion values are relative to mock siRNA, DMSO treated cells. **(G)** *Salmonella* invasion of WT and Δβ-PIX Hap1 cells pretreated with DMSO, IPA3, G5555 or PAK18. Invasion values are relative to those in DMSO treated WT cells. **(H)** Immunoblot confirming the knockout or knockdown of proteins in the indicated cell lines and conditions. Protein levels quantified from band intensities, relative to total tubulin, are displayed below the immunoblots. All Error bars indicate SD. NS–no significant difference, **—P <0.01, * P <0.05 (ANOVA followed by a post hoc Dunnett's comparison). (TIF)

**S5 Fig.** Invasion of WT *Salmonella* (20 mins) into MEF **(A)** and HeLa **(B)** cells expressing no construct (-) or β-PIX, invasion values are relative to corresponding no construct controls. **(C)** Expression of Full length and truncated β-PIX constructs confirmed by immunoblotting of corresponding cell lysates using a GFP antibody (Red), and a tubulin antibody (Green) as a loading control. **(D)** Invasion of WT *Salmonella* (20 mins) into WT and Δβ-PIX Hap1 cells, as well as Δβ-PIX Hap1 cells expressing one the various β-PIX constructs used in (Fig 5A), as indicated. Invasion values are relative to those in WT Hap1 cells. **(E)** Invasion of WT *Salmonella* into WT and Δβ-PIX Hap1 cells expressing the indicated construct: control (-) or β-PIX harbouring a mutation in its DH domain that blocks GEF activity (+ β-PIX $^{L238R,L239S}$). Invasion values are relative to those in control WT Hap1 cells. **(F)** Fluorescence microscopy images of Δβ-PIX cells expressing indicated Emerald-tagged β-PIX constructs (green). Actin stained with Texas Red Phalloidin (purple). Scale bars are 10 μm. **(G)** Fluorescence microscopy images of MEF and HeLa cells expressing Emerald-tagged β-PIX (green), infected with Alexa-fluor-350 stained WT *Salmonella* (10 mins). Actin stained with Texas Red Phalloidin (purple). Scale bars are 10 μm. **(H)** Invasion of WT *Salmonella* into WT, ΔPAK1 and WT Hap1 cells treated with IPA3 and pre-treated with either Mock or GIT1 siRNA (72 hrs prior to infection). Invasion values are relative to those in mock siRNA treated WT cells. All Error bars indicate SD. NS–no significant difference, **—P <0.01, * P <0.05 (ANOVA followed by a post hoc Dunnett's comparison). (TIF)

**S6 Fig. (A)** Fluorescence microscopy of EPEC attachment (90 mins) to WT, ΔPAK1 and ΔPAK1 Hap1 cell expressing PAK1$^{K299R}$ or PAK1$^{T423E}$, cells stained with Texas-red phalloidin to visualise actin (Red) and EPEC stained using an anti-intimin antibody (Green) Scale bar is 20 μm **(B)** Attachment of WT EPEC to WT and ΔPAK1 Hap1 cells expressing the indicated construct: control (-), kinase-dead PAK1 (PAK1$^{K299R}$) or constitutively kinase-active PAK1 (PAK1$^{T423E}$). Attachment values relative to control WT Hap1 cells. **(C)** Attachment of WT EPEC to WT and ΔPAK1 Hap1 cells expressing the indicated construct: control (-), the kinase domain of PAK1 (kinase), the PAK1 kinase domain fused to the PH domain of ARNO (PH-kinase), the CPP of PAK1 (CPP) or the CPP of PAK1 fused to the PH domain of ARNO (PH-CPP). All attachment values are relative to those in control WT Hap1 cells. **(D)** Fluorescence microscopy of EPEC pedestal formation (90 mins) to WT, ΔPAK1 and ΔPAK1 Hap1 cell expressing PAK1, PAK1$^{K299R}$, PAK1Δkinase or PAK1$^{T423E}$, cells stained with Texas-red phalloidin to visualise actin (Red) and EPEC stained using an anti-intimin antibody (Green). **(E)** Number of pedestals formed by EPEC per cell (90 mins) on the cells described in **(D)**. All Error bars indicate SD. NS–no significant difference, $^{**}$—P <0.01, $^{*}$ P <0.05 (ANOVA followed by a post hoc Dunnett's comparison).
(TIF)

**S7 Fig. (A)** Fluorescence microscopy of WT and ΔPAK1 Hap1 cells expressing Em-Rac1 (green) infected with alexa-fluor 350 stained WT Salmonella (blue– 10 mins). Actin stained with Texas-Red Phalloidin (purple) Scale bars are 10 μm **(B)** Invasion of WT *Salmonella* (20 mins) into WT, ΔPAK1 and WT Hap1 cells pretreated with IPA3. Cells were also pretreated with either DMSO (-) or the FAK kinase inhibitor (+PF573228). Invasion values are relative to those in WT control cells. **(C)** Invasion of WT *Salmonella* into WT, ΔPAK1 and WT Hap1 cells pretreated with IPA3. Cells expressed indicated constructs: Control (-) or FAK-related nonkinase (+ FRNK). Values are relative to those in control WT Hap1 cells. **(D)** Fluorescence microscopy of WT, ΔPAK1 and Δβ-PIX Hap1 cells expressing Em-Vinculin (green) infected with alexa-fluor 350 stained WT Salmonella (blue– 10 mins). Actin stained with Texas-Red Phalloidin (purple) Scale bars are 10 μm. All Error bars indicate SD. NS–no significant difference, $^{**}$—P <0.01, $^{*}$ P <0.05 (ANOVA followed by a post hoc Dunnett's comparison).
(TIF)

**S1 Data. Mass Spectrometry Data.**
(XLSX)

## Acknowledgments

We thank Rachael Stone and Nick Greene for technical assistance.

## Author Contributions

**Conceptualization:** Anthony Davidson, Joe Tyler, Peter Hume, Vassilis Koronakis.

**Data curation:** Anthony Davidson, Joe Tyler, Vikash Singh.

**Formal analysis:** Anthony Davidson, Joe Tyler, Vikash Singh.

**Funding acquisition:** Vassilis Koronakis.

**Investigation:** Anthony Davidson, Joe Tyler.

**Methodology:** Anthony Davidson, Joe Tyler, Vassilis Koronakis.

**Supervision:** Vassilis Koronakis.

**Validation:** Anthony Davidson, Joe Tyler.

**Visualization:** Anthony Davidson, Joe Tyler.

**Writing – original draft:** Anthony Davidson, Joe Tyler, Peter Hume.

**Writing – review & editing:** Anthony Davidson, Joe Tyler, Peter Hume, Vassilis Koronakis.

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
