## [Decision Letter · Decision Letter 0]

21 May 2021

Dear Dr. Koronakis,

Thank you very much for submitting your manuscript "Kinase-independent function of PAK is crucial for pathogen-mediated actin remodelling" for consideration at PLOS Pathogens. As with all papers reviewed by the journal, your manuscript was reviewed by members of the editorial board and by several independent reviewers. In light of the reviews (below this email), we would like to invite the resubmission of a significantly-revised version that takes into account the reviewers' comments. Note that we may send your manuscript back to the more critical reviewers upon resubmission

A particularly important point: You will note that all three reviewers asked the same question: If the role of PAK in invasion is independent of its kinase activity, why do all of the PAK inhibitors block invasion? How do these inhibitors act? Presumably most of them directly inhibit kinase activity? Please describe the panel of inhibitors in more detail, and address this question directly in the text.

Several other issues were raised by more than one reviewer:

1. Two of the reviewers also commented on the use of inappropriate statistical methods (t-tests vs. ANOVA). As some of your data are on the edge of statistical significance, a new analysis may cause you to temper some of your conclusions.

2. Two of the reviewers also commented on the lack of sufficient detail in your methods. Specifically, were transfections with PAK and PIX constructs transient or stable, and if transient how did you identify transfected cells?

3. Please supply a file containing the mass spectroscopy data, which should be made publicly available.

The above are the issues that were raised by more than one reviewer. However, all of the other reviewer comments are thoughtful and reasonable, and while they may require some additional experiments I suspect are readily addressable. When you are ready to resubmit, please be prepared to provide the following:

(1) A letter containing a detailed list of your responses to the review comments and a description of the

changes you have made in the manuscript.

(2) Two versions of the manuscript: one with either highlights or tracked changes denoting where the

text has been changed; the other a clean version (uploaded as the manuscript file).

We hope to receive your revised manuscript within 60 days or less. If you anticipate any delay in its

return, we ask that you let us know the expected resubmission date by replying to this email.

We cannot make any decision about publication until we have seen the revised manuscript and your response to the reviewers' comments. Your revised manuscript is also likely to be sent to reviewers for further evaluation.

Sincerely,

Jim Casanova

Guest Editor

PLOS Pathogens

Guy Tran Van Nhieu

Section Editor

PLOS Pathogens

Kasturi Haldar

Editor-in-Chief

PLOS Pathogens

orcid.org/0000-0001-5065-158X

Michael Malim

Editor-in-Chief

PLOS Pathogens

orcid.org/0000-0002-7699-2064

Reviewer's Responses to Questions

**Part I - Summary**

Reviewer #1: This manuscript by Davidson et al. addresses the question of how mammalian signaling pathways are manipulated by bacterial pathogens. The authors dissect the functions of PAK-family kinases and their associated proteins during Salmonella invasion and EPEC adherence. Beginning with chemical inhibitors of the PAKs, following-up with various host gene knockouts and knockdowns, and transitioning to mutant constructs for multiple signaling proteins, the authors executed an impressive repertoire of loss-of-function, dominant negative, rescue, and gain-of-function studies in cultured cells. Surprisingly, they found that the kinase activity of the PAKs was not required for pathogen colonization. Instead, signaling via small G-proteins, guanine nucleotide exchange factors, and polyproline motifs was important for efficient invasion/adherence. This paper thus provides remarkably detailed analyses of the host pathways that are exploited by Salmonella and EPEC. The only drawbacks are (i) that the methods underlying the phenotypic scoring are unclear, (ii) that several of the conclusions are based on borderline statistical analyses, and (iii) that the EPEC studies do not address actin rearrangements. If these aspects are cleared-up, the paper will be a nice contribution to the Salmonella, EPEC, and PAK kinase fields.

Reviewer #2: The authors examine role of PAK in Salmonella invasion. They propose an interesting model whereby PAK recruits B-PIX to invasion sites to stabilize actin dynamics and possibly alter focal adhesions elsewhere (this part of the model is not clear to me). They use a variety of molecular, biochemical and pharmacological strategies, and they also examine pathogenic E coli to see if their results are shared by another pathogen. Overall the data is interesting but I think premature in that it lacks important controls for the invasion assay used and the focal adhesion idea is not developed. There is also a concern about the finding that PAK inhibitors block Salmonella invasion, which does not seem to match the findings of the rest of the paper.

Reviewer #3: Davidson et al. have presented a study investigating the molecular mechanism of pathogen-driven modulation of host actin cytoskeleton and the role played by PAK. They reveal that when salmonella attach onto host cells, effector proteins stimulate group1 PAK. PAK1 interacts with β-PIX through the central poly-proline rich region (CPP) and inhibits its activity in a kinase independent manner to impede β-PIX activity on cytoskeleton reorganisation. Thus, bacteria and other pathogens modulate host cytoskeleton signalling through PAK1. The paper is well written in general although these are some issues with figure citation as detailed below.

Specific points

The statistical methods were not clearly stated/described and the clarification of using a particular statistical test is also missing. ANOVA should be used for three or more groups of data to find out the relationship or significance between independent and dependent variables.

Figure 1A: WT control could be carried out along with DMSO control to analyse the impact of DMSO in cells.

Figure 1B: Not clear in legend or methods if transfection is transient or stable. This question relates though out the paper – please evidence the level of expression that was achieved in transfection experiments – is it at endogenous levels or overexpression ?

Figure 1E: why was the S144 site selected? What happens at the PAK1 autophosphorylation site in the kinase domain upon pathogen binding? How does this result relate to later discussion of a kinase independent mechanism?

Figure 4D: Cited in text as interaction data but no present in the figure – please check in manuscript citations It would be good to see the Mass Spectroscopy data showing the proteins recruited by the PAK1 polyproline motif.

Figure 5 is not cited in the text but have assumed the first mentions of Figure 6 are actually Figure 5. – please check in manuscript citations

Figure 5D Authors suggest a mechanistic pathway for b-PIX as an inhibitor of Salmonella invasion by Rac1 mis localisation. Please can authors provide Rac1 localisation imaging in WT and Δ PAK1 cells to evidence this hypothesis.

The authors validated the PAK inhibitors experiment in WT MEF and HeLa cells. For knockout, siRNA, overexpression, and localisation experiments used Hap1 cells were used. In the later context it would be important to replicate the results using another cell line. By doing so it would offer the reproducibility of the experiments. This should be mentioned in the discussion.

**Part II – Major Issues: Key Experiments Required for Acceptance**

Reviewer #1: 1. The authors should clarify how the mutant rescue studies were performed. Aside from one sentence in the Methods, the DNA transfections were not described. Were the knockout cells transiently or stably transfected with the rescue constructs? If transient, how were the transfected cells identified? If stable, how was this accomplished? In either case, were the PAK1 and PIX mutants expressed at the predicted sizes? Ideally, their expression levels would be compared to endogenous PAK1 or PIX.

2. The authors draw several conclusions about the functionalities of PAK (and other) mutants based on fairly modest increases in relative invasion or adherence. However, a few of these conclusions are based on changes of 15-20% with t-test p-values close to 0.05. This is somewhat problematic, because the most appropriate statistical tests for the data depicted in the bar graphs would be ANOVAs (with post-hoc tests). Several conclusions (3ACD, 4DE, 5BC, 6D) might not hold up under these analyses. Which of the increases (e.g., from PAK K299R, PH-CPP, siPIX, and PIX mutant) are confirmed to be statistically significant? If they are not, then several conclusions should be softened.

3. The authors used EPEC to broaden the understanding of how pathogens exploit PAK signaling. But the focus on that part of the manuscript was on bacterial adherence rather than actin rearrangements. To draw the parallels between Salmonella and EPEC, some aspect of F-actin pedestal formation by EPEC should be quantified for the treatments in Fig.6. The acid wash retention measurement is not sufficient in this area. Several representative images analogous to those in Fig.1C, 2D, and S3 should also be incorporated into the main figure or a supplementary figure to support the conclusions.

Reviewer #2: Major comments

-Introduction: “Previous research has identified a role for PAK in the cell’s

nuclear response to Salmonella infection [20], as well as in the generation of an acute inflammatory

response[21]; however, no role for PAK in bacterial uptake has been demonstrated.” Prior studies by Chen et al (J Exp Med, 1999) showed that kinase-dead mutant of PAK does not affect Salmonella invasion. Proline mutants were also found to have no effect on invasion. This prior work should be cited early on, rather than the discussion.

-The abstract, intro and prior studies by Chen et al (J Exp Med 1999) all indicate PAK kinase activity is not required for Salmonella invasion. So why do PAK kinase inhibitors suppress Salmonella invasion in Figure 1?

-Same question for Figure 2 data… how can kinase dead PAK1 restore invasion if PAK kinase inhibitors block invasion? Am I missing something?

-The authors are using an unconventional invasion assay and need to provide controls, especially considering the fact that Hap1 cells are rarely studied for Salmonella infection. Importantly, are they measuring invasion or rather bacteria associated with the cell surface (i.e. outside)? Does cell surface binding change with altered PAK activity/expression?

-Where is the mass spectrometry data for His-PSB-CPP vs His-PSB alone? This is discussed in text but not shown anywhere.

-Figure 4. The interaction of PH-CPP construct with b-PIX is shown. What about full length PAK1… does it bind as well? A co-ip assay should be attempted. Does binding change during infection?

-Figure 5. Is b-PIX affecting recruitment of FA markers to Salmonella invasion sites?

- Figure 6. Authors indicate they are examining “attachment” – are they referring to EPEC pedestals (A/E lesion formation) or just passive binding to the cell surface? Images are required to show what they are measuring.

Reviewer #3: As mentioned above Authors suggest a mechanistic pathway for b-PIX as an inhibitor of Salmonella invasion by Rac1 mis localisation. Please can authors provide Rac1 localisation imaging in WT and Δ PAK1 cells to evidence this hypothesis.

**Part III – Minor Issues: Editorial and Data Presentation Modifications**

Reviewer #1: 4. How do the PAK inhibitors work? The peptide inhibitor PAK18 was described, but the mechanisms of actions of the others are unclear. I would’ve guessed that at least one of them targets kinase activity, but the authors’ results suggest that they affect other PAK functions. How do the specific actions of the inhibitors (e.g., IPA3 or G5555) fit with the model that the CRIB-CPP module can bypass their effects during Salmonella signaling?

5. In addition to the missing Methods regarding transfection/rescue described above, several other aspects of the Methods should be provided, including the concentrations of siRNAs, MOIs for Salmonella- and EPEC-HAP1 infections, and a full section on fluorescence microscopy.

6. The labels for mutants 83/86 and 107 in Fig.2A appear to be switched.

Reviewer #2: -Figure S2: The wild type PAK1 control should be added to this figure.

-Figure 3B would benefit from a column indicating “Restores invasion” similar to Figure 2B

Reviewer #3: (No Response)

PLOS authors have the option to publish the peer review history of their article (what does this mean?). If published, this will include your full peer review and any attached files.

Reviewer #1: No

Reviewer #2: No

Reviewer #3: No
---

## [Decision Letter · Decision Letter 1]

17 Aug 2021

Dear Vassilis, 

We are pleased to inform you that your manuscript 'A kinase-independent function of PAK is crucial for pathogen-mediated actin remodelling' has been provisionally accepted for publication in PLOS Pathogens.

Best regards,

Jim Casanova

Guest Editor

PLOS Pathogens

Guy Tran Van Nhieu

Section Editor

PLOS Pathogens

Kasturi Haldar

Editor-in-Chief

PLOS Pathogens

orcid.org/0000-0001-5065-158X

Michael Malim

Editor-in-Chief

PLOS Pathogens

orcid.org/0000-0002-7699-2064

The authors have done an excellent job of addressing reviewer concerns. There are only two very minor issues raised by Reviewer 1:

1. The legend to Fig. 5 needs to be changed to reflect the lack of statistical significance revealed after the updated statistical analysis.

2. Fig. 6 panels D, E. G - the appropriate statistical post-test here is Tukey's, not Dunnett's. Re-analysis is unlikely to change the authors' interpretation of the data, but should be done nonetheless.

Reviewer Comments (if any, and for reference):

Reviewer's Responses to Questions

**Part I - Summary**

Reviewer #1: The authors have adequately addressed my questions. It looks like their PAK1 antibody performs pretty crappily on immunoblots, but I commend them for their efforts in the supplement. This is a nice paper on a complex signaling mechanism.

My only other comments are that in Fig.5, one of the differences became NS so the p<0.05 in the legend is no longer applicable, and that in Fig.6DEG I think the most appropriate post-test for the internal comparisons would be the Tukey (Dunnett is only for comparisons to a single control mean). These are inconsequential comments, and the authors’ conclusions remain valid.

Finally, “depletion” seems to be misspelled as “depleteion” in the model figure (unless this is a special UK spelling).

Reviewer #2: The authors have addressed my comments.

**Part II – Major Issues: Key Experiments Required for Acceptance**

Reviewer #1: (No Response)

Reviewer #2: None

**Part III – Minor Issues: Editorial and Data Presentation Modifications**

Reviewer #1: (No Response)

Reviewer #2: None

PLOS authors have the option to publish the peer review history of their article (what does this mean?). If published, this will include your full peer review and any attached files.

Reviewer #1: No

Reviewer #2: No

---

## [Editor Report · Acceptance letter]

24 Aug 2021

Dear Prof. Koronakis,

We are delighted to inform you that your manuscript, "A kinase-independent function of PAK is crucial for pathogen-mediated actin remodelling," has been formally accepted for publication in PLOS Pathogens.

Best regards,

Kasturi Haldar

Editor-in-Chief

PLOS Pathogens

orcid.org/0000-0001-5065-158X

Michael Malim

Editor-in-Chief

PLOS Pathogens

orcid.org/0000-0002-7699-2064